# A genetic map of human metabolism across the allele frequency spectrum

Martijn Zoodsma [1,2], Carl Beuchel [1,2], Summaira Yasmeen [1], Leonhard Kohleick [1], Aakash Nepal[1], Mine Koprulu [3], Florian Kronenberg [4], Manuel Mayr [5], Alice Williamson [1,3,6], Maik Pietzner [1,2,3,7] & Claudia Langenberg [1,2,3,7]

Genetic studies of human metabolism have been limited in scale and allelic breadth. Here we provide a data-driven map of the genetic regulation of circulating small molecules and lipoprotein characteristics (249 traits) measured using proton nuclear magnetic resonance spectroscopy across the allele frequency spectrum in ~450,000 individuals. Trans-ancestral meta-analyses identify 29,824 locus–metabolite associations mapping to 753 regions with effects largely consistent between men and women and large ancestral groups represented in UK Biobank. We observe and classify extreme genetic pleiotropy, identify regulators of lipid metabolism, and assign effector genes at >100 loci through rare-to-common allelic series. We propose roles for genes less established in metabolic control (for example, *SIDT2*), genes characterized by phenotypic heterogeneity (for example, *APOA1*) and genes with specific disease relevance (for example, *VEGFA*). Our study demonstrates the value of broad, large-scale metabolomic phenotyping to identify and characterize regulators of human metabolism.

Our understanding of human metabolism is mostly based on dedicated hypothesis testing in experimental settings, informed by model organisms or observations in patients with rare diseases. Only recently has high-throughput profiling of small molecules in large-scale studies enabled systematic testing of genetic variation across the genome and provided an agnostic approach for discovering genes that encode key metabolic regulators[1–11]. These efforts have provided important new insights into how genetic variation shapes human chemical and metabolic individuality[1] and have corroborated a large body of biochemical knowledge[1,2,10,12].

The importance of such genome–metabolome-wide association studies (mGWAS) extends beyond the mapping of biochemical pathways, sometimes demonstrating almost immediate clinical value. They provided examples of how readily available supplementation strategies may prevent disease or delay onset in high-risk individuals,

such as serine for macular telangiectasia type 2, a rare eye disorder[2]. They further identified unknown variants that affect the absorption, distribution, metabolism and excretion of exogenous compounds, most importantly drugs[1,13], thereby providing pathways to mitigate adverse drug effects. However, there are several challenges that currently limit the potential of mGWAS analyses, particularly for causal inference. These include (1) the still rather small number of, at most, a dozen genetic variants linked to single molecules, (2) the inability to distinguish whether pleiotropic variants act on different molecules or pathways independently (horizontal pleiotropy), or whether they serve as 'root causes' of successive downstream changes (vertical pleiotropy), (3) the difficulty in distinguishing between locus-specific and metabolite abundance effects when colocalization at disease-risk loci is observed[1] and (4) the challenge of confidently assigning effector genes at newly identified loci.

[1]Computational Medicine, Berlin Institute of Health at Charité – Universitätsmedizin Berlin, Berlin, Germany. [2]DZHK (German Centre for Cardiovascular Research), partner site Berlin, Berlin, Germany. [3]Precision Healthcare Institute, Queen Mary University of London, London, UK. [4]Institute of Genetic Epidemiology, Medical University of Innsbruck, Innsbruck, Austria. [5]National Heart and Lung Institute, Imperial College London, London, UK. [6]Friede Springer Cardiovascular Prevention Center at Charité, Charité University Medicine Berlin, Berlin, Germany. [7]These authors contributed equally: Maik Pietzner, Claudia Langenberg. ✉e-mail: maik.pietzner@bih-charite.de; claudia.langenberg@qmul.ac.uk

Here, we integrated rare (based on whole exome sequencing) and common genetic variation with measures of 249 metabolic phenotypes, including small molecules and detailed lipoprotein characteristics, among >450,000 UK Biobank (UKB) participants representing three distinct ancestries. We demonstrate largely consistent genetic regulation across ancestries and sexes for almost 30,000 locus–metabolite associations and systematically categorize abundant genetic pleiotropy. By integrating machine-learning-derived effector gene assignments with rare exonic variation, we identify previously unknown regulators of metabolism and observe heterogeneity in association profiles for variants mapping to the same gene. Finally, we demonstrate how systematic integration of statistical colocalization and Mendelian randomization can identify pathways with the potential to mitigate cardiovascular disease (CVD) risk beyond current approaches focused primarily on lowering low-density lipoprotein (LDL) cholesterol.

## Results

We integrated genome-wide association studies (GWAS; population-specific minor allele frequency (MAF) ≥0.5%) with rare exome-wide association studies (ExWAS; MAF ≤0.05%) on plasma concentrations of 249 metabolite phenotypes, quantified using $^1$H nuclear magnetic resonance (NMR) spectroscopy. We included up to 450,000 UKB participants across three major ancestries (British White European, EUR ($n = 434,646$); British African, BA ($n = 6,573$); British Central/South Asian, BSA ($n = 8,796$)) (Extended Data Fig. 1). The NMR measures comprised 14 lipoprotein subclasses and associated characteristics (that is, extra-large very-low-density lipoprotein (VLDL) to small high-density lipoprotein (HDL) particles), along with small molecules such as amino acids and ketone bodies quantified in molar concentration units (Supplementary Table 1).

### Common genetic variation underlying circulating metabolites

We identified 29,824 regional sentinel–NMR measure associations in trans-ancestral meta-analyses, representing 753 nonoverlapping genomic regions (Fig. 1a and Supplementary Table 2). Nearly half of these regions ($n = 359, 47\%$) associated with more than ten NMR measures, demonstrating considerable pleiotropy. Characteristics of large HDL particles, such as particle size and lipid composition, were associated with the largest number of regions (median 166, interquartile range 126–195), compared with all NMR measures (median 105, interquartile range 68–142), findings that considerably extended previous work[3] and replicated parallel efforts using UKB[9] (Extended Data Fig. 2). Genes with well-characterized roles in human metabolism were significantly enriched across different significance bins (adjusted $P$ values $<4.24 \times 10^{-9}$; Supplementary Fig. 1), suggesting that ever-larger studies of omnigenic traits, such as metabolites, still yield biological plausible findings.

We observed significant evidence of heterogeneity ($P < 1 \times 10^{-4}$) across ancestries for very few loci ($n = 342; 1.14\%$), and ancestral-wise comparison of effect estimates demonstrated largely concordant effect estimates (Fig. 1c,d, Extended Data Fig. 3 and Supplementary Table 3). All sentinels seen in individuals of British African and British Central/South Asian ancestry were replicated in individuals of European ancestry, except for one locus that was specific to British Africans. The previously reported[14] missense variant rs3211938 within *CD36*, which is common among individuals of African ancestry ($MAF_{BA} = 0.12$) but absent among individuals of European ancestry ($MAF_{EUR} = 0.0$), was significantly associated ($P$ values $<1.49 \times 10^{-9}$) with lower plasma concentrations of omega 3 fatty acids and 15 other NMR measures, including lipoprotein particle characteristics. This is in line with the role of *CD36* encoding for a fatty acid translocase, facilitating the recognition and uptake of long-chain fatty acids. We note that the sample sizes in the smaller ancestral groups did not permit comprehensive replication.

### Sex-differential effects at loci encoding metabolic genes

While we observed highly correlated effect sizes across female and male participants (median $r = 0.98$, range 0.90–0.99), we also identified 360 putative sex-differential loci for 239 NMR measures, representing 1,800 heterogenous associations in sex-stratified meta-analyses (heterogeneity $P$ value $<5 \times 10^{-8}$), most of which (65.3%; $n = 1,175$ loci) could not be explained by confounding factors (Supplementary Note, Supplementary Fig. 2 and Supplementary Table 4). Putative sex-differential loci were generally directionally concordant between the sexes (Fig. 2a), in line with previous proteomics analyses and suggesting that significant sex interactions do not reflect sex-discordant effects[15].

### Refinement of regional associations through multi-ancestry fine-mapping

We next used a two-stage strategy to refine regional associations to a smaller number of candidate causal variants. We first identified 3,007 statistically independent metabolite quantitative trait loci (mQTLs) associated with one or more NMR measure, representing a total of 43,322 credible set–NMR measurement pairs (Supplementary Table 5). Lead fine-mapped mQTLs per NMR trait explained on average 6.9% (range 0.57–13.42%) of variance in plasma metabolite concentrations (Extended Data Fig. 4). Second, we leveraged the different linkage disequilibrium (LD) structure in British African and British Central/South Asian individuals to further refine 3,386 credible sets that contained >1 variant and with suggestive evidence in either ancestry, leading to an increase in the number of credible sets with high-confidence variants and decrease in mean credible set size from 9 to 4 variants (Supplementary Note and Supplementary Fig. 3). Trans-ancestral fine-mapping improved resolution in loci that did not resolve in individuals of European ancestry alone, but we note that the overall improvement was marginal. Instead of refining already tight credible sets, future studies should therefore focus on scaling discovery in non-European ancestries to identify unknown causal variants.

### Biological reclassification of established 'lipid' loci

To assess the value of metabogenomic studies of $^1$H NMR-spectrometry-based lipoprotein profiling over standard clinical markers, we classified NMR metabolome association profiles for 1,657 genetic variants reported for commonly measured clinical markers (LDL cholesterol, HDL cholesterol, total cholesterol and triglycerides) obtained in 1.6 million people[16]. Around 25% of associated variants had the corresponding NMR measure among the top 10% of the most strongly associated NMR measures, with 22.5% of genetic variants showing significantly stronger association with refined lipoprotein measures compared with their matching measure on the NMR platform, an observation most pronounced for non-HDL and LDL cholesterol concentrations (Fig. 2b). Relevant loci for lipoprotein metabolism can thus be discovered using readily available clinical measurements; however, refined lipoprotein profiles are necessary for better understanding the relevant biological pathways, including any inference about druggability or use for genetic causal inference methods. One such example was the *PNPLA3* locus (tagged by rs3747207, associated with LDL cholesterol by the Global Lipids Genetics Consortium; $\beta = -0.014, P = 2.3 \times 10^{-21}$), where we observed no association with LDL cholesterol ($\beta = -0.001, P = 0.49$) but with LDL particle size ($\beta = 0.045, P = 1.04 \times 10^{-73}$), and multiple characteristics of extra-large VLDL particles (Extended Data Fig. 5). The intronic rs3747207 variant is in strong LD ($r^2 = 0.98$) with the well-known missense variant rs738409 (p.Ile148Met) that has been demonstrated to confer hepatic lipid accumulation by altering ubiquitination of patatin-like phospholipase domain-containing protein 3 (PNPLA3)[17]. Our results provide human genetic support for a recently proposed role of PNPLA3 in the secretion of large VLDL particles[18].

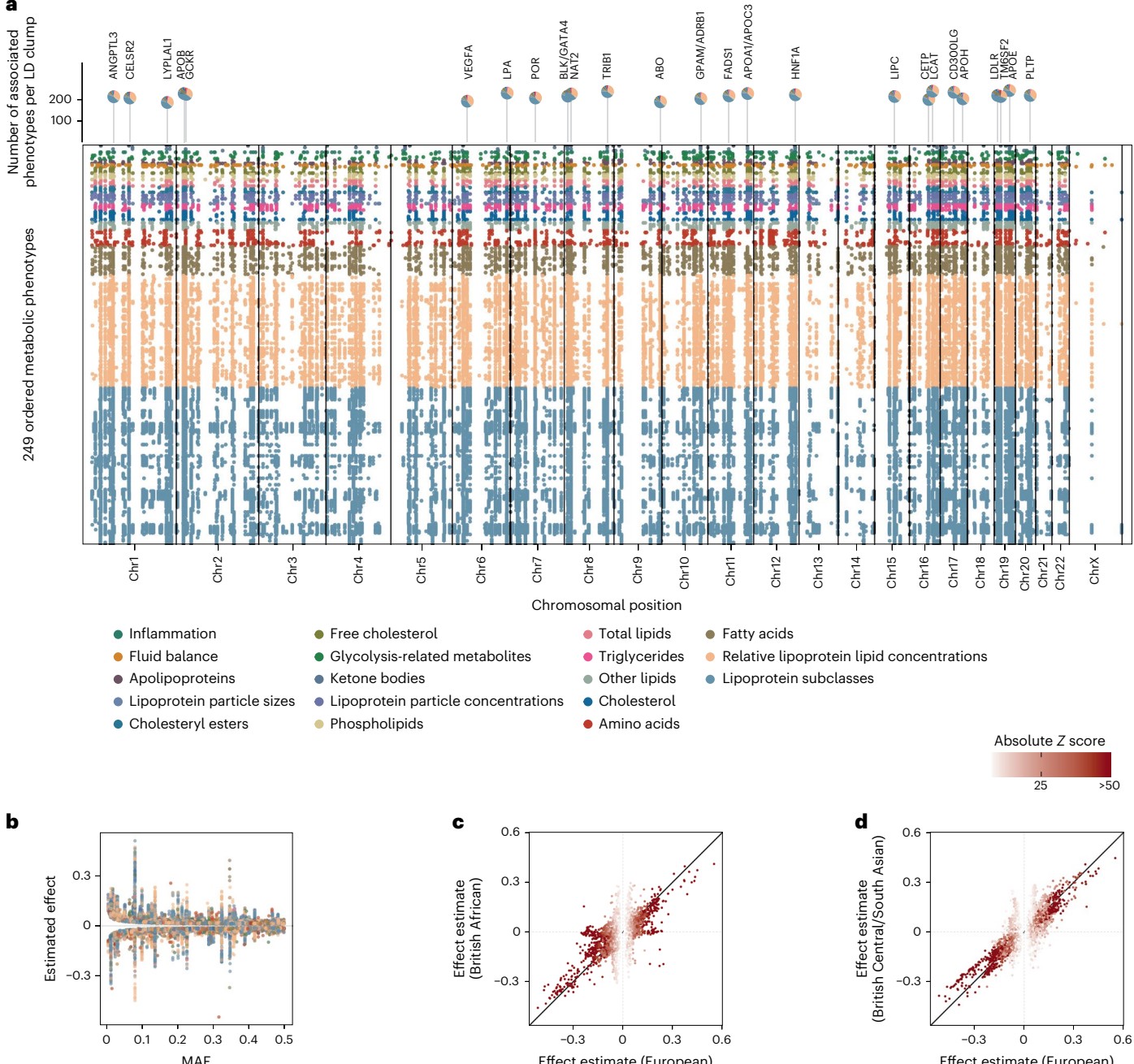

**Fig. 1 | Common genetic regulation of circulating metabolites. a**, A top-down Manhattan plot showing trans-ancestral sentinel variants for 249 metabolic phenotypes at a metabolome-adjusted genome-wide significance threshold of $P < 2.0 \times 10^{-10}$. Each row represents an NMR measure, colored for biochemical class. Chromosomal positions are shown on the $x$ axis. $P$ values are raw $-\log_{10}(P \text{ value})$ from a two-sided $Z$ test across effect estimates derived within three ancestral groups. **b**, Weighted average allele frequency compared with estimated effect size for trans-ancestral sentinel variants. Points are colored for biochemical classification. **c**, A comparison of effect sizes between British White European samples ($x$ axis) and British African samples ($y$ axis). We considered variants that were significant in either population. **d**, Similar to **c** but comparing British Central/South Asian samples. Dots are colored according to their absolute $Z$ score in British White European samples.

## Machine-learning-guided effector gene assignment

We successfully assigned effector genes for almost three-quarters of European ancestry fine-mapped mQTLs (73.6%; $n = 2{,}213$) with at least moderate confidence (candidate gene score ≥1.5, range 0–3), including about 28.2% with high-confidence assignments (score ≥2; $n = 848$), by training a machine learning model that integrates functional genomic resources with pathway information inspired by the ProGeM framework[19] (Supplementary Table 6). For example, we prioritized the fatty acid elongase gene *ELOVL6* for 16 different VLDL/HDL characteristics

(tagged by rs3813829). The gene product, ELOVL fatty acid elongase 6, catalyzes the rate-limiting step in long-chain fatty acid elongation, which are subsequently incorporated into lipoprotein particles. We also prioritized genes with upstream roles in metabolism, including a locus on 17q25.3 where we prioritized cytohesin-1 (*CYTH1*) as the putative effector gene for 5 independent genetic variants linked to 11 distinct NMR measures mostly comprising characteristics of VLDL particles. *CYTH1*, previously associated with type 2 diabetes[20], promotes activation of ADP-ribosylation factors (ARF)1, ARF5 and ARF6, regulators of

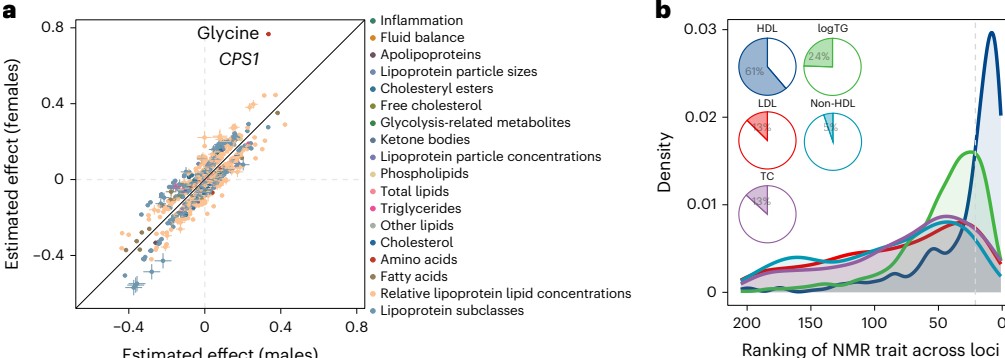

**Fig. 2 | Putative sex-differential loci and reclassification of established lipid loci. a**, Comparison of effect sizes of putatively sex-differential loci (defined as loci with heterogeneity $P < 5 \times 10^{-8}$ in a two-sided $Z$-score meta-analysis across the sexes). **b**, Rank distributions for each of the five matching NMR traits compared with the Lipids Genetics traits across genetic loci. Per locus–trait combination, 205 lipid-related NMR traits were ranked based on their absolute effect size and compared with the NMR trait that corresponds the Lipids Genetics consortium trait. Pie charts show the percentage of loci where the corresponding NMR trait is ranked among the top 10% of associated traits. TC, total cholesterol; TG, triglycerides.

lipid vesicle transport, membrane lipid composition and modification[21], demonstrating a relevant but indirect link to lipoprotein metabolism.

We observed considerable overlap of machine-learning-guided effector gene predictions (top three genes) with those reported based on manually curated biological plausibility (191 out of 283 loci)[3] or based on colocalization with protein quantitative trait loci (pQTLs) that have not been used to train the algorithm[22] (81 out of 143; Supplementary Table 6). While missing overlap indicates room for improvement, 24 high-confidence assignments strongly disagreed with either external source (gene score > 2 but no match among pQTLs prioritized or manually curated ones). For example, we prioritized *PEPD* (score 2.42) as opposed to *CEBPA*[3] for rs62102718. PEPD encodes peptidase D, which has been shown to promote adipose tissue fibrosis in mouse knock-out models promoting insulin resistance[23]. Insulin resistance, in turn, provides a very plausible explanation for the pleiotropic effect of the variant on diverse lipoprotein characteristics ($n = 31$).

### Tissue distribution of effector genes
Assigned effector genes were significantly enriched in different tissues, reflecting known and lesser-established organ contributions (Extended Data Fig. 6a and Supplementary Table 7). Genes characteristic of the liver, adipose tissue, adrenal gland and female breast tissue (probably reflecting its high adipose tissue content) were significantly enriched among effector gene sets across the metabolic measures captured by NMR. This included significant enrichment of all amino acids in liver tissue (for example, phenylalanine: odds ratio (OR) 14.8, $P < 1.3 \times 10^{-8}$, histidine: OR 7.9, $P < 2.9 \times 10^{-11}$) but also for skeletal muscle in alanine metabolism (OR 3.82; $P < 7.9 \times 10^{-9}$). Similar enrichments were observed when using the closest gene instead of our annotated effector genes for mQTLs (Extended Data Fig. 6b).

### Metabolic versus systemic pleiotropy
Pleiotropy is widespread but poorly understood. We developed a framework to characterize four different modes of metabolic pleiotropy (Fig. 3a–d, Extended Data Fig. 7, Supplementary Table 6 and Methods). About half of the pleiotropic mQTLs ($n = 880$; ≥2 NMR measures) showed evidence for two different modes of vertical pleiotropy. First, within confined pathways ($n = 218$; 'pathway pleiotropy'; Fig. 3a) or, second, as a function of the correlation with the 'lead' NMR measure ($n = 662$; 'proportional pleiotropy'; Fig. 3b). A prototypical example for proportional pleiotropy was an mQTL tagged by rs624698 for which we prioritized *ANGPTL3* as the likely effector gene (Fig. 3b). Angiopoietin-like 3, encoded by *ANGPTL3*, inhibits lipoprotein lipase activity but also endothelial lipase, resulting in increased triglycerides, HDL cholesterol and

phospholipid concentrations, consistent with HDL-particle characteristics being the most strongly associated NMR measure ($P < 1.0 \times 10^{-546}$). Other associations reflected downstream effects on lipoprotein metabolism rather than acting on independent pathways (Fig. 3b), considerably expanding previous genetic observations[24].

The remaining half of pleiotropic mQTLs showed evidence for two modes of horizontal pleiotropy: those with evidence for 'disproportional pleiotropy' ($n = 68$) and a larger group with evidence for 'nonspecific pleiotropy' ($n = 720$). For example, a small deletion on chromosome 1 (chr1:92982441:CA>C) was associated with a highly correlated cluster of NMR measures, including characteristics of intermediate density lipoprotein (IDL), LDL and VLDL particles (Fig. 3c), but for which we detected no correlation of association strengths according to the lead NMR measure, the concentration of esterified cholesterol in medium-sized VLDL particles ($P < 6.8 \times 10^{-14}$). We prioritized *EVI5* as the most likely effector gene, supported by previous studies on rare functional variants[25]. The gene product of *EVI5*, ectopic viral integration site 5, has no apparent link to (lipoprotein) metabolism, in line with most of the gene assignments for mQTLs with a similar nonspecific pleiotropy pattern. An example of nonspecific pleiotropy was the *APOB* missense variant rs676210 (p.Pro2739Leu) associated with 126 NMR measures across the entire lipoprotein density range, but also creatinine and glycoprotein acetyl concentrations (Fig. 3d). The differential effects of the same genetic variation on distinct lipoprotein subgroups aligns with changes in lipid profiles seen with mipomersen, an antisense oligonucleotide against *APOB*, that demonstrated reductions in LDL cholesterol but also subsequent increases in the triglyceride content of VLDL particles as hepatic adaption occurs[26].

Modes of molecular pleiotropy only partially translated into phenotypic pleiotropy (Fig. 3e,f). We observed a twofold enrichment of 'proportional pleiotropic' (OR 2.11; $P < 2.0 \times 10^{-14}$) and to a lesser extent an enrichment of 'nonspecific pleiotropic' (OR 1.52; $P < 1.1 \times 10^{-5}$) variants among variants reported in the GWAS Catalog for ≥5 nonmetabolomic trait categories (Methods). By contrast, the set of pleiotropic GWAS Catalog variants was significantly depleted for 'specific' mQTLs (OR 0.42; $P < 1.6 \times 10^{-21}$). Systemic mechanisms explaining effects of 'proportional' and 'nonspecific' pleiotropic mQTLs were further indicated by a more than 20-fold significant enrichment of associated trait categories such as 'metabolic disease', 'fatty liver disease' and 'arterial disorders' (Fig. 3g).

### Convergence of common and rare genetic variation shaping metabolism
We next sought to understand convergence of rare and common genetic findings to systematically identify allelic series that increase confidence

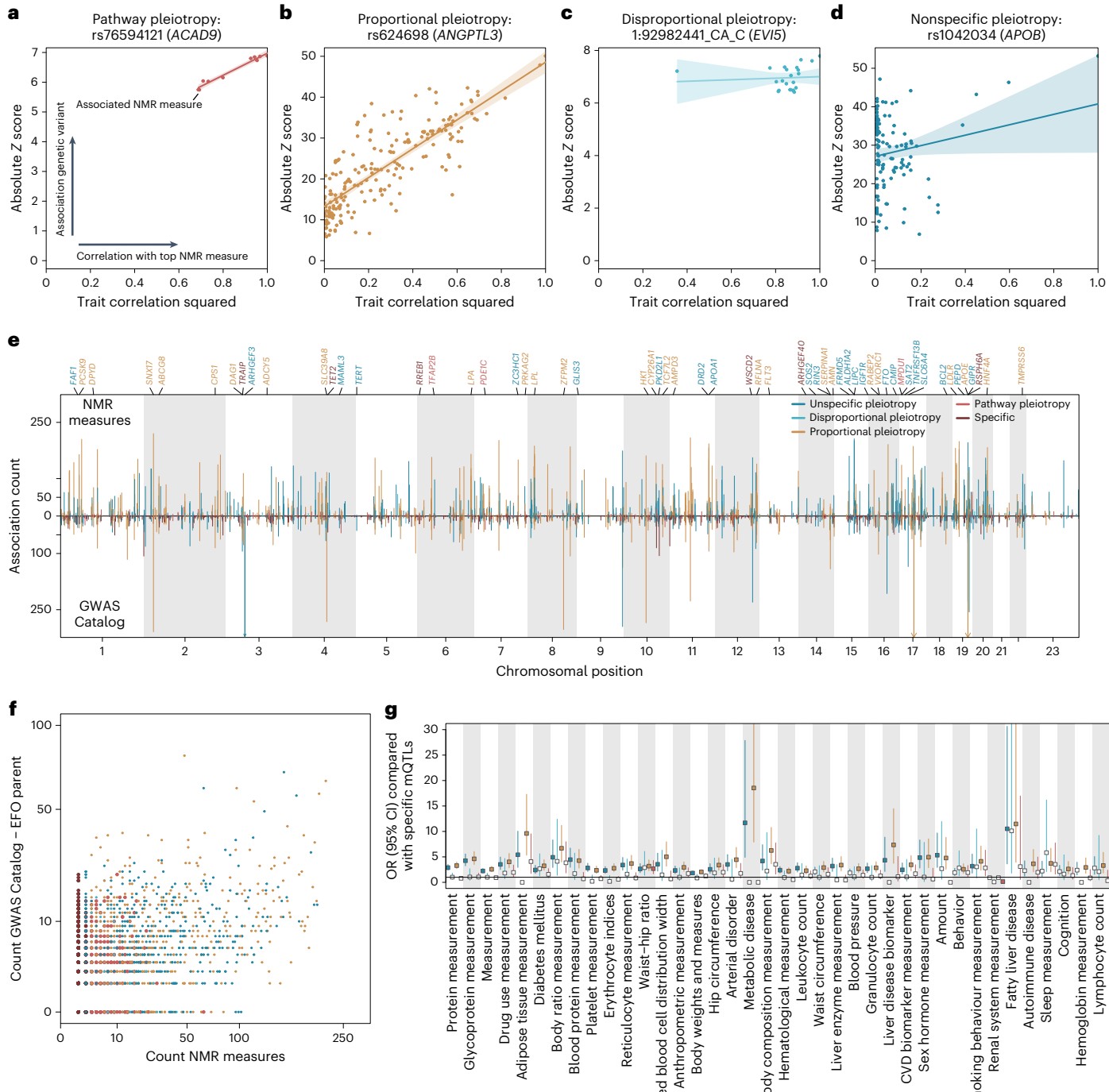

**Fig. 3 | Modes of pleiotropy. a–d,** Representative scatterplots opposing the squared trait correlation of the lead NMR measure for the listed variant against the absolute $Z$ score from linear regression models for all associated NMR measures. The colors indicate different modes of pleiotropy and correspond to the legend in **e**. For each plot, a linear regression fit (lines) with 95% confidence interval (bands) is given. Scatterplots in **a–d** represent examples of mQTLs classified as pathway pleiotropy (**a**), proportional pleiotropy (**b**), disproportional pleiotropy (**c**) and nonspecific pleiotropy (**d**). **e,** The number of associated NMR measures for each of 3,007 mQTL groups opposed to associations reported in the GWAS Catalog after pruning the GWAS Catalog for metabolic phenotypes (Methods). Coloring is according to modes of pleiotropy. **f,** A scatterplot opposing the number of associated NMR measures ($x$ axis) of each mQTL group with the number of reported EFO parent categories in the GWAS Catalog. **g,** ORs (rectangle) and 95% confidence intervals (CIs; lines) from logistic regression models testing whether EFO categories ($x$ axis) are more frequently reported for pleiotropic mQTL groups compared with specific ones. Darker colors indicated estimates passing corrected statistical significance. $n = 3,007$ mQTL groups have been used for enrichment testing.

in causal gene assignment. We identified rare variation (MAF ≤0.05%) in 209 genes to be significantly ($P < 1.1 \times 10^{-8}$) linked to one or more of 249 NMR measures combining ultrarare gene burden analysis (3,709 significant associations; Supplementary Table 8) and rare exonic variant analysis (4,131 significant associations; Supplementary Table 9).

Effect sizes were significantly larger compared with more frequent variant effects (Fig. 4a). For example, participants carrying rare predicted loss-of-function (LoF) variants in *SLC13A5* had more than 1.4 s.d. units higher plasma citrate concentrations per copy of the possibly damaging allele ($\beta = 1.41$; $P < 2.6 \times 10^{-20}$).

We also observed considerable pleiotropy, including 47 genes associated with 20 or more NMR measures. Many of these genes encode for well-known enzymes and transportes, with nearly half ($n = 23/51$ genes) being involved in (peripheral) cholesterol metabolism (Extended Data Fig. 8). Some rare pleiotropic variants with large effect sizes (MAF <0.02% and $\beta > 0.6$ s.d. units) pointed toward less-established regulators of metabolism, including *SIDT2* (chr11:117186662:C>T, $n = 124$ associated NMR traits), *JAK2* (chr9:5073770:G>T (p.Val617Phe), $n = 73$ associated NMR traits) or *CEP164* (chr11:117356670:C>G, $n = 49$ associated NMR traits). Experimental work already suggested a role for the gene product of *SIDT2* (SID1 transmembrane family member 2) in hepatic lipid metabolism and apolipoprotein A1 (ApoA1) secretion, the main protein component of HDL particles, which constituted the majority of associated NMR measures[27,28] (Fig. 4b). Variation in *JAK2* predisposes to somatic mutations inducing hematopoiesis of indeterminate potential (CHIP)[29], but other studies linked the gene product Janus kinase 2 (JAK2) to metabolism in liver[30], adipocytes[31] or macrophages[32]. The strong inverse association with parameters of HDL particles thereby best aligned with a role of JAK2 in promoting the interaction with ATP-binding cassette transporter A1 (*ABCA1*) and subsequent HDL-mediated lipid removal from cells, including atherogenic macrophages[32]. These findings considerably expanded an earlier hypothesis that attributed effects of the same *JAK2* variant on LDL cholesterol primarily to myeloid cells in a mouse model[33]. This hypothesis only partially aligns with—and in some respects contrasts—our human genetic findings across the lipoprotein-density gradient.

We observed strong overlap between gene burden and common variant findings, with 85.4% of rare variant ($n = 3,528$) and 75.5% of gene burden ($n = 2,802$) associations being <100 kb away from the nearest statistically independent lead credible set variant (Fig. 4c). By contrast, most common variant findings (92.3%) were not within 500 kb of matching rare variant/burden evidence. Notably, 12.1% of gene burden results were more than 1 Mb away from the next common credible set variant for the respective NMR measure, aligning with recent observations that both approaches prioritize partly different genes[34].

At 116 genes (55.5%), rare variant and/or burden evidence overlapped with effector gene predictions for close by common credible set variants (≤200 kb) for one or more associated NMR measure (Fig. 4d), providing independent support for allelic series (Fig. 4d and Supplementary Table 10). For example, we identified an allelic series composed of seven rare LoF, one gain-of-function and four common variants for serum citrate levels at *SLC13A5* encoding a sodium-dependent citrate co-transporter. Another allelic series at *ANKH* comprised four common variants (rs185448606, MAF 1.3%; rs17250977, MAF 4.0%; rs826351, MAF 44.3%; rs2921604, MAF 45.9%) and a rare missense variant chr5:14745916:T>C (MAF 0.0069%) being also associated with lower serum concentrations of citrate ($\beta = -2.18$ s.d. units, $P < 5.2 \times 10^{-11}$) (Fig. 4d). *ANKH* encodes a multipass transporter, recently shown to transport citrate[35], with an important role in bone health[35].

## Phenotypic heterogeneity within allelic series

We observed evidence that genetic variants within 17 genes associated with >10 NMR measures had differential metabolic consequences within an allelic series (Supplementary Table 10). The most outstanding example included seven variants (five rare; two common) and a cumulative burden of rare predicted LoF variants at *APOA1*. They distinctively associated with one or more of 87 NMR measures, most strongly with diverse characteristics of HDL particles of which the gene product, Apolipoprotein A1 (ApoA1), is the major component (Fig. 4e,f). This included four rare missense variants (MAF ≤0.03%) encoded in exon 4 that partly differentially associated with the number, size and cholesterol content of HDL particles (Fig. 4e), only one of which (p.Leu158Pro) primarily associated with serum ApoA1 concentrations and HDL particle number, mimicking the cumulative burden of high-confidence predicted LoF variants in *APOA1* and suggesting a potentially dysfunctional protein that lacks interaction with lecithin cholesterol acyl transferase to facilitate cholesterol uptake[36]. By contrast, p.Lys131del and p.Arg201Ser seemed to rather predispose to a shift in cholesterol content from large towards small HDL particles, a pattern opposed by p.Asp113Glu (Fig. 4e). Consistently, amyloid formation by ApoA1 has been observed in early case reports of p.Lys131del (ApoA-I_{Helsinki}[37]) in which HDL-cholesterol or ApoA1 concentrations are only mildly changed but aggregation of misfolded ApoA1 protein can confer organ damage later in life[38]. Because p.Asp113Glu and p.Arg201Ser have not yet been identified to cause amyloidosis, we cannot rule out the possibility that each variant maps to distinctive parts of ApoA1 with subsequently different consequences on function and/or stability (Supplementary Fig. 4). While results for serum ApoA1 concentrations were largely confirmed using an alternative assay, we observed some discrepancies that may imply that, in the presence of rare missense variants, the procedure to quantify ApoA1 concentrations from $^1$H NMR spectra may need recalibration.

## Phenotypic consequences of rare variation in metabolic genes

We observed a >3-fold enrichment of genes previously linked to Mendelian diseases[39] ('OMIM genes') among those associated with NMR measures in gene burden and rare exonic variant analyses (OR 3.30, $P < 6.5 \times 10^{-17}$; Supplementary Table 11), in line with previous mGWAS[1,2,7,8]. For 15 out of 106 genes, we found evidence of significantly associated disease risk ($P < 7.5 \times 10^{-7}$), largely replicating signs and symptoms of corresponding rare disorders (Supplementary Note and Supplementary Table 12). When we tested more generally whether a rare variant burden in metabolic genes was associated with disease susceptibility, we observed a significant enrichment among susceptibility genes for endocrine and metabolic disorders, such as type 2 diabetes and different lipidemias but not among other disease categories (Supplementary Fig. 5).

## Risk mitigation of atherosclerotic CVD beyond LDL cholesterol

Genetic predisposition to high LDL cholesterol is strongly associated with increased atherosclerotic CVD (ACVD) risk ('level effect'), and

**Fig. 4 | Rare coding variation associated with NMR measures and convergence with common variant associations. a**, Effect estimates against MAF of significantly associated gene burden (diamonds; two-sided $P < 1.2 \times 10^{-8}$ and rare exonic variants (MAF <0.05%; circles; two-sided $P < 2.0 \times 10^{-10}$). **b**, Effect estimates and two-sided raw $-\log_{10}(P$ values) for associations of the rare intronic variant chr11:117186662:C>T within *SIDT2* across all 249 NMR measures. The dotted horizontal line indicates the multiple testing threshold ($P < 2.0 \times 10^{-10}$). **c**, Genomic distance between gene burden (blue) or rare exonic variants (orange) toward the next common credible set variant. **d**, Evidence for allelic series based on (i) gene burden analysis (bottom), (ii) rare exonic variants (middle) and (iii) common variants with prioritized effector gene matching to the evidence from exonic analysis. For each gene, only the NMR measure most significantly associated with the strongest common variant is shown in cases where multiple NMR measures were associated. Some bars for the number of associated rare

exonic variants have been capped to fit into plotting margin, but the number is given in the plot. **e**, Effect estimates (dots) and 95% CIs (lines) from our European-based exWAS for 7 variants mapping to *APOA1* as well as a cumulative burden of high-confidence pLOF variants within *APOA1* and bespoke circulating measures of ApoA1 (clinical indicates measurements by immunoturbidimetric analysis on a Beckman Coulter AU5800) and HDL particles (color gradient). **f**, Top: a heatmap of standardized effect estimates (per variant) across 87 NMR measures for each associated variant and a cumulative burden within *APOA1*. Variants mapping into the region encoding the protein are surrounded by a rectangle. Variant effects have been aligned to the minor allele. Middle: the corresponding variants mapped to their respective transcripts encoding different forms of *APOA1*. Bottom: missense variants mapped onto the amino acid sequence of the protein. Variant names colored similarly had highly correlated association profiles.

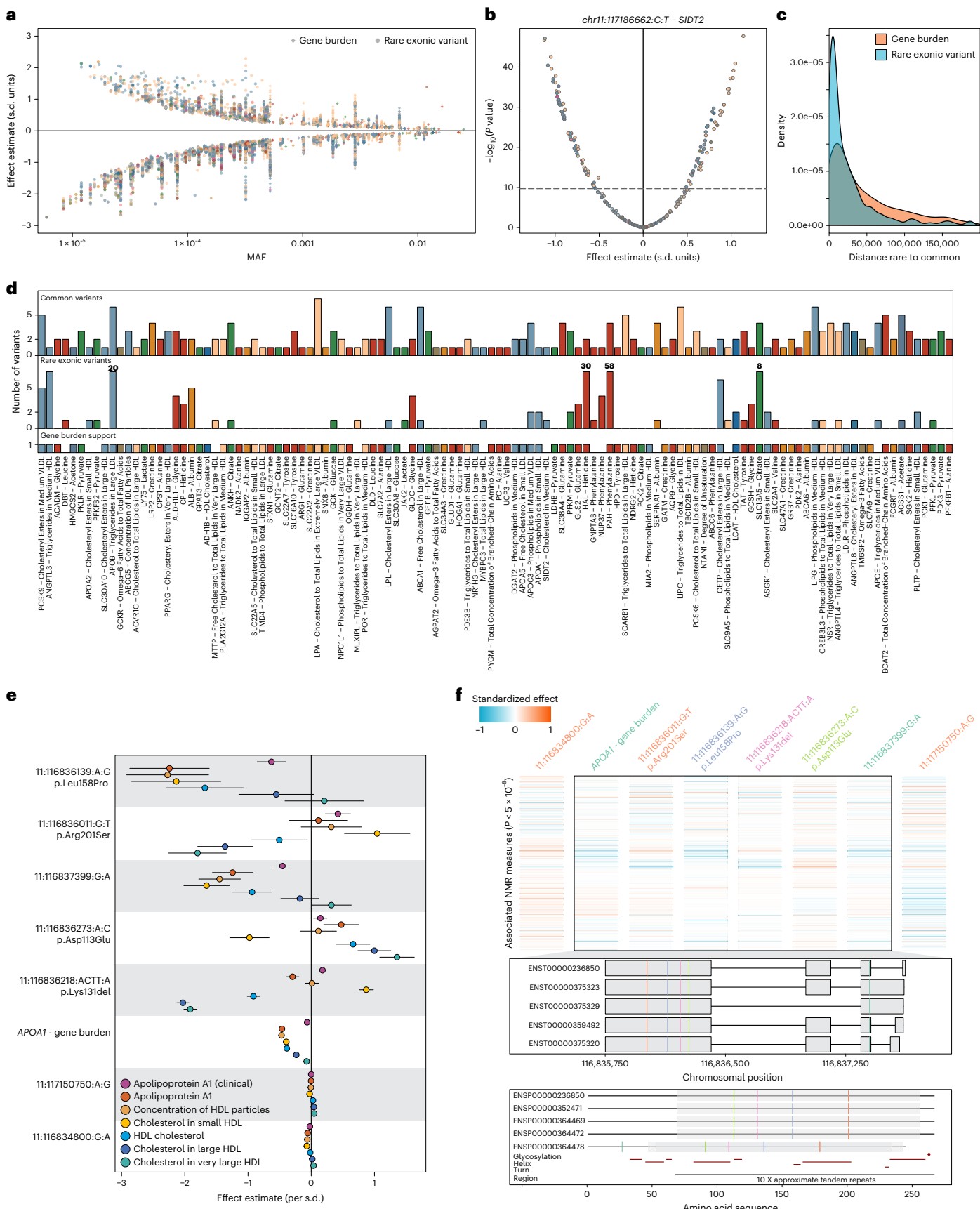

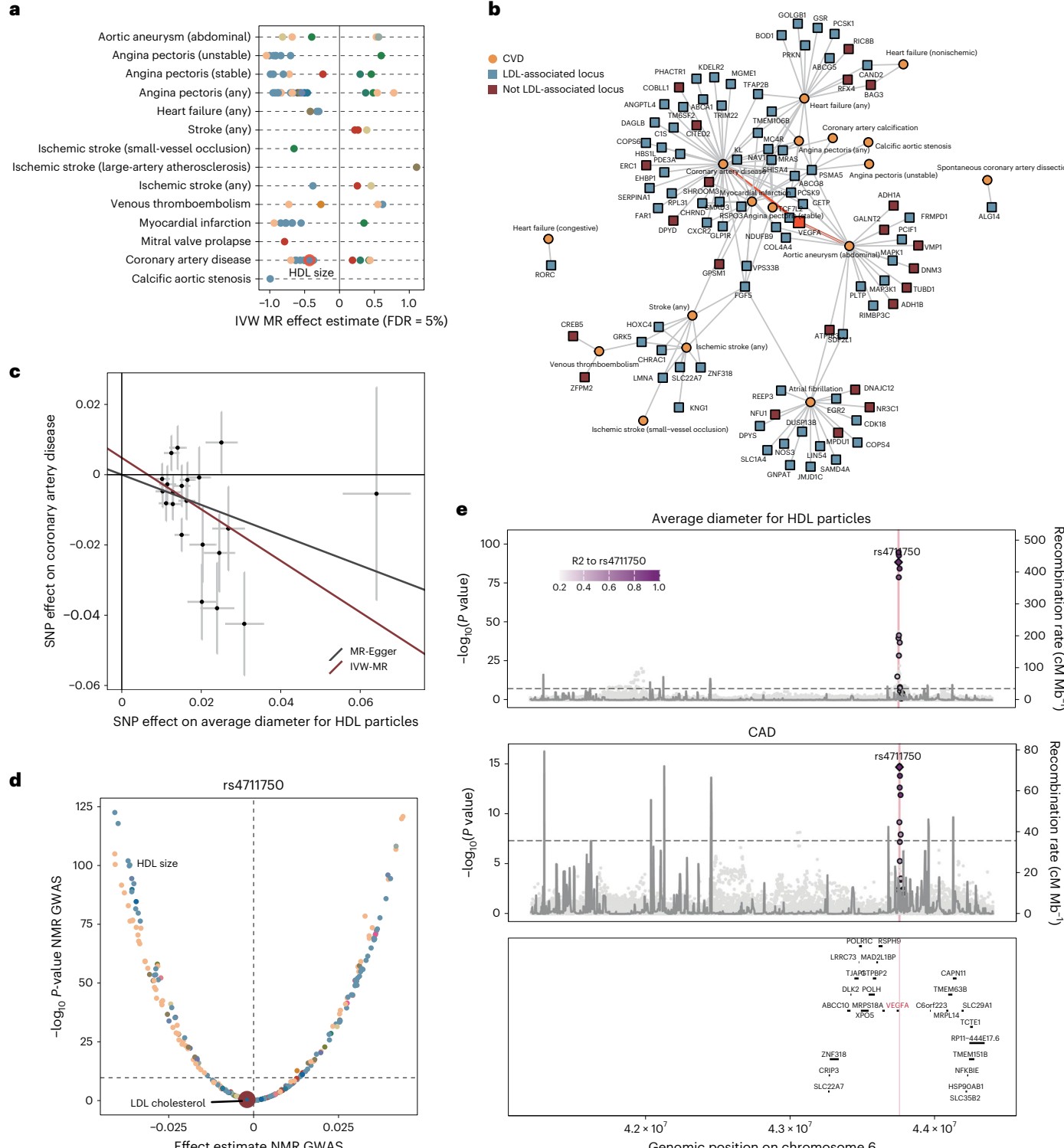

**Fig. 5 | Genetic prioritization to target residual cardiovascular risk. a,** Summary of two-sample Mendelian randomization analysis testing for putatively causal effects of NMR measures on the risk on diverse CVD. Shown are effect estimates for NMR–disease pairs passing multiple testing. Metabolites are colored according to the scheme from Fig. 1. **b,** Locus–disease network highlighting loci for which at least 1 NMR measure showed evidence of colocalization with 1 or more CVDs (PP ≥80%). Only loci without evidence for unspecific pleiotropy are depicted. Loci were annotated with the most likely causal gene. Loci colored in blue showed evidence for being associated with LDL cholesterol whereas red did not. **c,** Dose–response plot for SNPs associated with

HDL particle size (after filtering for pleiotropic SNPs) against the risk for CAD. Effect estimates (dots) and 95% CIs (lines) are given and MR-regression lines added. Effect estimates derived from our European ancestry-based GWAS ($x$ axis, $n = 434,646$) and Aragam et al.[42] ($y$ axis, $n = 1,589,012$). **d,** Effect of rs4711750 across the NMR metabolome. The $y$ axis is a two-sided raw $-\log_{10}(P \text{ value})$ derived from the European ancestry-based GWAS ($n = 434,646$). **e,** LocusZoom plot centered around *VEGFA* demonstrating colocalization for the genetic signal for HDL particle size and CAD. The $y$ axis represents the raw $-\log_{10}(P \text{ value})$ from the European ancestry-based GWAS.

genetic variations that mimic potent drug targets, such as at *PCSK9*, show strong evidence of shared effects on both LDL cholesterol and ACVD ('locus effect')[40]. To identify potential pathways to mitigate the residual risk not addressed by lowering of LDL cholesterol[41], we systematically integrated outcome data across 25 CVD phenotypes[42–56] with NMR phenotypes (Supplementary Table 13).

We identified significant evidence (false discovery rate (FDR) <5%) for 1,146 'level effects' across 218 NMR measures with one or more of 22 CVD phenotypes using pleiotropy-curated genetic instruments in Mendelian randomization (Fig. 5a and Supplementary Table 14). Independently, we observed evidence for 5,527 'locus effects', suggesting a shared genetic architecture (posterior probability (PP) >80%) between 87 mQTLs associated with 247 NMR measures and 17 CVD phenotypes (Fig. 5b and Supplementary Table 15). For 46 NMR–CVD combinations, we found converging evidence for level and locus effects, including 23 not associated in our study with parameters of LDL metabolism (Fig. 5b), providing potential alternatives for addressing residual cardiovascular risk (Supplementary Table 16).

For example, we observed robust evidence that, among other measures related to HDL size and composition, genetic susceptibility to larger HDL particle size was associated with a 35% reduced risk of coronary artery disease (CAD; OR 0.65; 95% CI 0.50–0.83; $P_{adj}$ < 0.007; Fig. 5c) along with evidence of a shared and directionally concordant genetic signal at the *VEGFA* locus (rs4711750, PP 99%; Fig. 5e). The locus has been implicated in CAD risk[42], and our results now suggest that one likely pathway to modulate CAD risk might be via HDL particle size or characteristics of large HDL particles not captured by HDL cholesterol. Vascular endothelial growth factor A (VEGFA), encoded by *VEGFA*, is primarily known for its role in angiogenesis[57] but has been described as a regulatory factor of transendothelial transport of esterified cholesterol from HDL but not LDL particles via activation of scavenger receptor BI (SR-BI) during reverse cholesterol transport[58]. Inhibition of VEGFA is a major pharmaceutical target to suppress vascularization of malignant tumors[57], and agents targeting VEGF signaling are well known for adverse cardiovascular effects[59], suggesting that VEGFA activation, rather than inhibition, might be necessary to reduce CAD risk. Our observations contribute to a growing body of evidence that more tailored approaches, rather than increasing HDL cholesterol content, will probably be needed for potential cardiovascular benefits, given the discouraging trials for most agents increasing HDL cholesterol[60]. We note, however, that HDL-particle size might still only be a 'measurable' surrogate, rather than being the true underlying mechanism. For example, inhibition of reverse cholesterol transport via dysfunctional SR-BI increased HDL particle size as well as CAD risk[61].

### Disease-wide Mendelian randomization screen for nonlipoprotein measures

Having established pleiotropy categories, we finally aimed to demonstrate its application for nonlipid NMR measures in a disease-wide Mendelian randomization screen (Supplementary Note and Supplementary Table 17).

We observed converging evidence for a risk-increasing effect of genetically predicted plasma glycoprotein acetyl concentrations on type 2 diabetes risk (OR per 1 s.d. increase 1.67; $P < 3.9 \times 10^{-7}$) that persisted after exluding variants with evidence for phenotypic pleiotropy (OR 1.69; $P < 9.1 \times 10^{-5}$). This is in line with the rare LoF variant chr20:44413714:C>T (MAF 0.02%) within *HNF4A* on plasma glycoprotein acetyl concentrations ($\beta = 0.60$; $P < 8.3 \times 10^{-15}$) and the cumulative effect of ultrarare LoF *HNF4A* variants on type 2 diabetes risk (OR 2.68; $P = 6.5 \times 10^{-10}$). However, we note that plasma glycoprotein acetyl concentrations proxy a complex chronic inflammatory state[62] that warrants further follow-up analysis to establish mechanistic links to type 2 diabetes.

## Discussion

The genetic basis of circulating metabolites provides insights into the complexity of human metabolic regulation and its subsequent influence on health and disease. By integrating common and rare genetic variation with circulating metabolite concentrations in 450,000 individuals from three different ancestries, we provide here a data-driven map of the circulating metabolome across the allele frequency spectrum. This map identifies previously unrecognized modulators of metabolism with potential health implications.

By combining machine-learning-guided common variant-to-gene annotation with rare exonic variation, we provided high-confidence effector gene assignments at >100 loci, including some with less established roles in (lipoprotein) metabolism, such as *SIDT2*, presenting compelling candidates for functional follow-up studies in humans. Large-scale studies similar to ours, but with a broader coverage of the plasma metabolome, will probably uncover more genes with yet undefined roles in metabolism, complementing hypothesis-driven research in experimental models.

After more than two decades of GWAS, it has become clear that pleiotropic effects of genetic variants are ubiquitous (see, for example, ref. 63). Little distinction has been possible beyond the generic concepts of 'vertical' and 'horizontal' pleiotropy or measures of simple counting. We refine these concepts by observing variants associated with dozens of NMR measures but consistent with the concept of effects diluting or propagating along. Conversely, we observe variants associated with comparatively few NMR measures in an inconsistent pattern, suggesting distinct effects on otherwise highly correlated traits. Our data-driven approach augments previous concepts based on biochemical pathways reporting directionally discordant pleiotropy to discover metabolic bottlenecks[64].

Disturbance in metabolism or rearrangements thereof are a hallmark of many diseases, including those not classically considered as 'metabolic', such as eye disorders[2], but whether these are pathways for prevention or intervention, rather than a consequence of the disease, often remains elusive in humans. We demonstrated considerable overlap between mQTLs with disease risk loci, including rare-to-common allelic series that can reveal unknown effector genes. However, many such 'locus effects' were characterized by nonspecific pleiotropy, implicating the plasma metabolite as a bystander rather than cause of the disease. This observation aligns with the relatively few notable exceptions, such as HDL particle characteristics and CAD, from two-sample Mendelian randomization (MR) analyses that contrasted the broad spectrum of observed disease associations described for the same NMR platform[65]. These observations might be best explained by the concept of metabolic flexibility, which includes built-in redundancy in key pathways to combat various intrinsic and extrinsic perturbations.

An important distinction of our study compared with most previous efforts was the availability of highly standardized measurements in a well-designed single large cohort, mitigating influences of preanalytical variables and enabling analyses of even ultrarare variants. However, this also meant that we had little opportunity to investigate the influence of different states of metabolism on our genetic results (such as an overnight fast) or investigate robustness of findings in different environments or at scale in other ancestries. For example, UKB participants were not asked to fast overnight before their baseline visit, which has been shown to impact genetic findings[3]. Other limitations included the sensitivity and coverage of the [1]H NMR platform, and future efforts are likely to reveal more diverse phenotypic consequences of genetically constrained flexibility of human metabolism. Another technical aspect to consider in the interpretation of our results is the indirect nature of [1]H NMR derived measurements of certain analytes, including apolipoproteins, that may no longer be reliable in the presence of rare damaging variants that change the properties of apolipoproteins as observed for ApoA1.

## Online content

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

## Methods

### Study design

The UKB is a prospective cohort study from the UK that contains more than 500,000 volunteers between 40 and 69 years of age at inclusion. The study design, sample characteristics and genotype data have been described elsewhere[66,67]. The UKB was approved by the National Research Ethics Service Committee North West Multi-Centre Haydock and all study procedures were performed in accordance with the World Medical Association Declaration of Helsinki ethical principles for medical research. We included 460,036 individuals across the three major ancestries in UKB in our analyses for whom inclusion criteria (given consent to further usage of the data, availability of genetic data and passed quality control (QC) of genetic data) applied. Data from UKB were linked to death registries and hospital episode statistics (HES). We used the ancestry assignments as defined by the pan-UKB[68] and further assigned unclassified individuals to their respective ancestries based on a $k$-nearest neighbor approach using genetic principal components. All analyses were conducted under UKB applications 44448 and 30418.

### Metabolomic measurements

Up to 249 targeted metabolomic measurements were quantified using the Nightingale NMR platform in human EDTA plasma samples. Detailed experimental procedures for the NMR platform are described elsewhere[65,69]. The NMR platform covers a wide range of metabolic biomarkers, including lipoprotein lipids, fatty acids and small molecules such as amino acids, ketone bodies and glycolysis metabolites, quantified in molar concentration units. We combine here three data releases that cover the full breadth of the UKB. Metabolomics data were available for 482,276 individuals, including 19,699 samples with data from baseline and repeat visit.

Metabolites were reliably detected, with only one biomarker over 2.5% missingness in releases 1/2 (creatinine) and release 3 (3-hydroxybutyrate). Ninety-eight percent of the samples had <5% missingness over all biomarkers in releases 1/2 and release 3. We used the ukbnmr[70] R package (v2.2, R v4.3.2) for QC and removal of technical variation in the NMR data. This includes technical confounders such as sample preparation time, shipping plate well, spectrometer effects, time drift within spectrometers and outlier plates.

We removed samples that were flagged by Nightingale for poor quality and used the MICE (Multivariate Imputation by Chained Equations)[71] R package to impute the remaining dataset. In total, we imputed 0.16% and 0.17% of data in releases 1/2 and release 3, respectively.

We observed overall good consistency with the overlapping routine blood biomarkers previously measured in the same cohort (median $r$ = 0.9, range 0.62–0.94) (Extended Data Fig. 9).

### Adjustment of metabolomic data for medication use

We sought to adjust the NMR data for medication use, especially cholesterol-lowering medication, to avoid false-positive results driven by medication use in downstream genetic analyses. For male and female participants separately, we fit linear models to quantify the impact of six drug categories on each NMR phenotype: cholesterol-lowering medicine, blood pressure medication, diabetic medication including Metformin usage, oral contraceptive pill or minipill (female only) and hormone replacement therapy (female only) (UKB fields 6177 and 6153) (Supplementary Fig. 6 and Supplementary Table 18).

We used data from individuals with both baseline ($NMR_{baseline}$) and repeat ($NMR_{follow-up}$) assessment metabolic data available and estimated the effect of medication (med terms) in individuals that did not take any drugs at the time of the baseline visit ($n$ = 6,312 male, $n$ = 6,713 female participants) using the following model:

$$NMR_{baseline} \sim NMR_{follow-up} + age + BMI$$
$$+ med_{cholesterol} + med_{diabetic} + med_{contraception} + med_{hormone} + error.$$

We note that the sample sizes for diabetic medication ($n_{male}$ = 45, $n_{female}$ = 29), oral contraceptive medication ($n$ = 27) and hormone replacement therapy ($n$ = 148) were too small to reliably estimate any effects. Effect estimates for diabetic medication were correlated to estimates for cholesterol-lowering medicine. The effect estimates for blood pressure medication were minimal across the phenotypes. We considered thus only the impact of cholesterol-lowering medicine and corrected the metabolic data in a sex-specific manner.

### Genotyping and GWAS analyses

GWAS was performed on 249 metabolic traits measured by the NMR platform on British European ($n$ = 434,646), British Central/South Asian ($n$ = 8,796) and British African participants ($n$ = 6,573) that had complete phenotypic, covariate and genetic information available. We used the Haplotype Reference Consortium-imputed genetic data, including all autosomal chromosomes and the X chromosome. We performed GWAS under the additive model using REGENIE (v3.2.5)[72] that uses a two-step procedure to account for population structure. We derived a set of high-quality genotyped variants per population by applying the following filters: (MAF >1%, minor allele count (MAC) >100, missingness rate <10%, $P_{HWE} > 1 \times 10^{-15}$). Furthermore, linkage disequilibrium pruning was performed using a 1,000-kb window, shifting by 100 variants and removing variants with LD ($r^2$) >0.8. We used these variants as input for the first step of REGENIE to generate individual trait predictions using the leave-one-chromosome-out scheme. These predictions are used in the second step where individual variants are tested. Models were adjusted for age, sex and the first ten genetic principal components. We tested variants with a MAF >0.5%, amounting to 11.5 million variants in British European individuals, 11.5 million variants in British Central/South Asian individuals and 19.3 million variants in British African individuals.

For initial discovery, we performed a meta-analysis across the three ancestral groups using METAL[73]. We required variants to be present in at least two ancestral groups. To declare significance, we considered a stringent $P$-value threshold ($2.0 \times 10^{-10}$) by dividing the standard genome-wide threshold by the number of metabolic phenotypes ($5.0 \times 10^{-8}/249$).

We tested our results for genomic inflation and calculated the single-nucleotide polymorphism (SNP)-based heritability using LD-score regression[74] (Supplementary Table 19).

### Regional clumping and fine-mapping

We used regional clumping (±500 kb) around sentinel variants from the analyses including British European samples to select independent genomic regions associated with a metabolic phenotype and collapsed neighboring regions using BEDtools (v2.30.0). We treated the extended MHC region (chr6: 25.5–34.0 Mb) as one region.

Within each region of interest, excluding the MHC region, we performed statistical fine-mapping for all phenotypes associated with that region using the 'Sum of single effects' model (SuSiE) implemented in the susieR (v0.12.35) R package[75]. In brief, SuSiE uses a Bayesian framework for variable selection in a multiple regression problem with the aim to identify sets of independent variants each of which probably contains the true causally underlying genetic variant. We implemented the workflow using default prior and parameter settings, apart from the minimum absolute correlation, which we set to 0.1. Because SuSiE is implemented in a linear regression framework, we used the GWAS summary statistics with a matching correlation matrix of dosage genotypes instead of individual-level data to implement fine-mapping (susie_rss()) as recommended by the authors[75].

To determine the appropriate number of credible sets within each region, we iterated over the maximum credible sets parameter in susieR from two to ten, thus generating fine-mapped results constrained to a range of maximum number of credible sets. For each collection of credible sets, we pruned sets where the lead variant was correlated to the

lead variant of other credible sets ($r^2 > 0.25$). After pruning, we considered the fine-mapped results with the largest number of credible sets.

We performed several sensitivity analyses by computing joint models per locus–phenotype combination, jointly modeling the effect of all distinct lead credible set variants in a single linear model. Subsequently, we retained only credible sets where the lead variant reached genome-wide significance ($P = 5.0 \times 10^{-8}$) in both marginal and joint statistics. Furthermore, we ensured the estimated coefficients were directionally concordant and of similar magnitude between joint and marginal models (±25%). Linear models were implemented in R using the glm() function and used only unrelated British European participants and the same set of covariates as described above.

Finally, we used LD clumping ($r^2 > 0.6$) to identify credible sets shared across metabolic phenotypes.

We computed the correlation matrix with LDscore v2.0 using genetic data from 50,000 randomly selected, unrelated White European UKB participants. In situations where SuSiE did not deliver a credible set, we used the Wakefield approximation[76] to compute 95%-credible sets.

### Replication of genetic associations

We replicated our trans-ancestral genetic signals using two independent studies: (1) the so-far largest published mGWAS[3] and (2) a parallel effort using overlapping UKB data[9], both using the same NMR platform. We considered a set of metabolic traits that were directly measured by the NMR platform and not inferred from other traits to avoid multiplicative errors in these more sensitive phenotypes. In total, we were able to match 144 (Karjalainen et al.[3]) and 169 (Tambets et al.[9]) metabolic traits, for which we compared sentinel variants that passed metabolome-adjusted, genome-wide significance in our trans-ancestral meta-analysis and that overlapped between the studies.

### Causal gene assignment

To assign candidate genes for all metabolite QTLs residing outside the MHC region, we first collected annotations for each genetic variant or proxies thereof ($r^2 > 0.6$), including distance to the gene body and putative functional consequences based on the Variant Effect Predictor (VEP) tool offered by Ensembl. We further collated up to ten closest genes within a 2-Mb window and subsequent gene features such as: (1) eQTL evidence for a given variant–gene pair for each tissue available in the eQTL Catalogue release 7[77]; (2) evidence of being annotated as metabolic in the MGI or Orphanet databases as defined in ProGem[19]; (3) evidence of being listed in the Online Mendelian Inheritance in Man (OMIM) database[39]; (4) and evidence of being an already assigned drug target in Open Targets[78] clinical stages III and IV.

With no universally accepted standard for variant-to-gene assignments, we relied on prior biological and genomic information to create three sets of 'putative true positive' (PTP) set: genes part of cholesterol pathway in the Kyoto Encyclopedia of Genes and Genomes (KEGG)[79] or REACTOME[80] database ($n = 6,791, 722$ unique SNPs), lipid pathway ($n = 5,670, 603$ unique SNPs) and amino acid-related pathway ($n = 8,349, 895$ unique SNPs). We used all fine-mapped SNPs associated with metabolites classified in the respective NMR metabolite class (Cholesterol: cholesterol, cholesteryl esters, free cholesterol; Lipid: total lipids, other lipids, relative lipid concentration, phospholipids; Amino Acid: amino acid) in the PTP set and used overlapping SNPs in only one PTP set. We trained (7:3 training:test ratio without overlapping variants) a random forest classifier using fivefold cross-validation with subsampling to account for the unbalanced datasets (scikit-learn v1.4.1). We used the balanced accuracy score to choose the best-performing forest from each training set. Subsequently, we used the best-performing classifier from each PTP set to assign candidate scores for all putative effector genes across the entire set of metabolite QTLs. We calculated the median score across classifiers and selected the highest-scoring gene per variant. Within each PTP set, we omitted features used to define true positive sets. Each of the three classifiers exhibited consistent performance (mean ROC-AUC: 0.80, mean balanced accuracy score 0.69) (Supplementary Fig. 7). We used the sum across all three classifiers to assign effector gene scores but present only genes as potential effector genes that reached sufficient support as indicated by largest difference between consecutively prioritized genes.

To provide another layer of evidence for assignment of causal genes at metabolic loci, we performed *cis*-colocalization with protein targets measured in the independent Fenland study[22]. *Cis* (for example, gene body ± 500 kb) summary statistics were preprocessed using MungeSumStats[81]. To relax the single causal variant assumption, we used a colocalization approach where we fine-mapped all traits with SuSiE and then performed colocalization among all credible sets using functionality of the coloc (v5.2.3)[82,83] and susieR (v0.12.35)[75] R packages. For this, we set the prior probability that a SNP is associated with both traits to $5 \times 10^{-6}$ and restricted the maximum number of credible sets for the outcome data to five[82].

### Tissue enrichment of metabolic loci

We tested whether genes proximal to metabolic loci and assigned effector genes were enriched in tissue compartments by leveraging data from the Human Protein Atlas[84]. Specifically, we used a two-sided Fisher's test whether metabolic genes were enriched among tissue-specific genes (tissue-enriched or tissue-enhanced as defined by the Protein Atlas) against all protein-coding genes as background.

### Pleiotropy assignment and overlap with the GWAS Catalog

To assign modes of pleiotropy for each mQTL, we first clumped lead credible set variants across NMR measures by LD, collating variants with $r^2 \geq 0.6$ as a single signal, referred to hereafter as mQTL group. This was done based on dosage files of all unrelated British European UKB participants and implemented with the igraph (v.2.0.1.1) package in R. For each mQTL, we computed pairwise Pearson correlation coefficients among associated NMR measures. We classified each mQTL group on: (1) the 25th percentile of all pairwise correlations, and (2) the Pearson correlation coefficient between the association strengths for each measure ($-\log_{10}(P \text{value})$) and its correlation coefficient with the most strongly associated measure within the mQTL. The latter is a measure to what extent the association between NMR measures at a given locus ('pleiotropy') can be explained by being correlated with the most proximal associated measure. Based on opposing those two measures for all mQTLs we defined the following five groups: (1) 'specific' mQTLs associated with only ≤3 highly correlated NMR measures (rho ≥0.6); (2) 'pathway pleiotropic' mQTLs associated with highly correlated NMR measures (rho ≥0.6) that followed the described association pattern (rho ≥0.6); (3) 'proportional pleiotropic' mQTL groups associated with, in part, uncorrelated NMR measures but highly correlated association statistics (rho ≥0.6); (4) 'disproportional pleiotropic' mQTLs associated with highly correlated NMR measures (rho ≥0.6), but without evidence that this translated into a correlation of association statistics (rho <0.6), and; (5) all remaining mQTLs as 'unspecific pleiotropic' groups.

To quantify the extent to which our pleiotropy assignment extends beyond the NMR measures analyzed here, we intersected mQTLs and proxies thereof with results reported in the GWAS Catalog (downloaded 20 May 2024). We first pruned GWAS Catalog entries for those with mapped traits (to minimize double counting), results that met genome-wide significance ($P < 5 \times 10^{-8}$) and had location information available. We further dropped results similar to NMR measures based on broad Experimental Factor Ontology (EFO) terms (for example, EFO:0005105 and child terms indicating 'lipid or lipoprotein measurement'). To further account for traits mapping to similar categories, we iteratively traced back-mapped EFO terms to broader parent terms. We finally classified mQTLs to be 'specific' in the GWAS Catalog if they associated with fewer than five parent EFO terms and 'unspecific' otherwise.

## Integration with cardiovascular endpoints

We next aimed to investigate the shared genetic basis of the 249 NMR and 25 selected CVD traits. We utilized public databases (GWAS Catalog, openGWAS, CVD-KP) to collect CVD data comprising the largest currently publicly available GWAS datasets on CAD and myocardial infarction, angina pectoris, aortic aneurysm, heart failure and stroke, and peripheral arterial disease, including two to five subtypes for some phenotypes (Supplementary Table 13). Data were harmonized and, if necessary, lifted over to GRCh37 using the MungeSumstats (v1.13.2) R package[81]. We queried mQTL lead variants and proxies in strong LD ($r^2 > 0.8$; LD backbone based on UKB, as described above) of each NMR trait in each region and corresponding summary statistics for each CVD trait.

To investigate 'locus' effects, we performed statistical colocalization for all combinations of the NMR traits–CVD traits as described before (see 'Causal gene assignment' section).

To estimate 'level' effects of NMR metabolite concentrations on CVD outcomes, we performed Mendelian Randomization analysis using the TwoSampleMR package (v0.5.1), implementing the inverse-variance weighted and the MR-Egger methods. We used all 249 NMR metabolites as exposure variables, the 25 CVDs as outcome variables and assessed separately four sets of instruments: (1) sentinel variants, (2) lead credible set variants, (3) lead credible set variants restricted for molecular pleiotropy (for example, 'pathway pleiotropy') and (4) lead credible set variants restricted for both molecular and phenotypic pleiotropy. We used the Wald ratio method to estimate the effect of NMR concentrations on CVD outcomes using only single genetic variants[85]. We used MR-Egger to test for evidence of a pleiotropic association, an intercept $P$ value >0.0001 indicating evidence of no pleiotropy and checked for concordance between the effect estimates of inverse-variance weighted Mendelian randomisation (IVW-MR), MR-Egger and single genetic variant MR. We controlled the FDR at 5% (ref. 86). To further limit the possible extent of pleiotropic associations, we only reported 'level effects' passing these filters in the variant sets 2–4, prioritizing the association in the more stringent variant set.

The overlap of 'locus effects' showing no 'disproportional pleiotropy' according to the section 'Pleiotropy assignment and overlap with the GWAS Catalog' as well as a significant single variant MR (FDR 5%) and 'level effects' calculated from metabolite-specific or metabolite- and phenome-specific variants was used to identify gene–metabolite pairs associated with CVD risk independent of LDL metabolism. We considered loci as independent from LDL metabolism if they did not associate with clinical LDL cholesterol at the locus with $P < 2.0 \times 10^{-10}$ and the effect estimate of any variant on clinical LDL-C ranked upward the 80th percentile of all effect estimates at the locus.

## Whole exome sequencing data QC for rare variant analyses

An in-depth description of whole exome sequencing, including experimental details, variant calling and standard QC measures for the UKB has been extensively reported by Backman et al.[87] We performed additional QC steps at the UKB Research Analysis Platform (RAP; https://ukbiobank.dnanexus.com/).

We used bcftools (v1.15.1) to process population-level Variant Call Format (pVCF) files. Initially, we normalized the data using the reference sequence GRCh38 build, followed by splitting multiallelic variants. Subsequently, we conducted QC on these variants using a set of parameters outlined below to filter high-quality variants for downstream genetic analyses. Genotypes for SNPs were set to missing if the read depth was less than 7 (or less than 10 for INDELs) or if the genotype quality was below 20. Furthermore, we excluded variants if the allele balance was less than 0.25 or greater than 0.8 in heterozygous carriers. Finally, we excluded variants with missingness >50%.

## Variant annotation and gene burden masks

Variants were annotated using ENSEMBL VEP[88] (v106.1) with the most severe consequence for each variant chosen across all protein-coding transcripts. We further utilized additional plugins REVEL[89], CADD v1.6[90] and LOFTEE[91] for variant annotation. Based on these scores, we defined six partially overlapping variant masks: (1) high-confidence predicted LoF (pLOF, based on LOFTEE and includes stop-gained, splice site disrupting, and frameshift variants); (2) any pLOF assigned high impact by VEP; (3) pLOF and high-impact missense variants (CADD score >20 or REVEL score >0.5); (4) pLOF and any missense variants; (5) only high-impact variants; and (6) any missense variants but not pLOF. We tested synonymous variants separately as a negative control. We tested each mask in different MAF bins, using 0.5% and 0.005% as thresholds.

We performed rare variant association testing (RVAT) using whole exome sequencing (WES) data across 249 NMR phenotypes using REGENIE (v3.1.1) via the DNAnexus Swiss Army Knife tool (v4.9.1). Similar to common variant GWASs, we used a two-step approach by REGENIE. We additionally generated step 1 leave-one-chromosome-out (LOCO) files with and without adjusting for common signals via a polygenic score (PGS derived from all lead credible set variant per NMR trait) in the RVAT models per phenotype. All RVAT models were then adjusted for PGS in addition to age, biological sex, fasting duration and the first ten genetic PCs. We first performed aggregated gene burden testing across for 19,026 genes using a set of masks as defined above. For gene burden testing, we used the aggregated Cauchy association test to estimate $P$ values for each gene across masks and allele frequency bins. The aggregated Cauchy association test first computes $P$ values for all sets defined by various masks within a gene and then takes these $P$ values as input to compute one $P$ value for the respective gene via a well-approximated Cauchy distribution.

We performed single variant association testing for exonic variants (ExWAS). For the ExWAS, we tested variants with MAC >5 and reported results for variants with MAF <0.0005. We have performed these analyses in individuals of British European, British African and British Central/South Asian ancestry.

We considered findings as robust if they passed multiple-testing-corrected statistical significance (gene burden: $P < 1.2 \times 10^{-8}$ (corrected for the number of genes × number of traits); ExWAS: $P < 2.0 \times 10^{-10}$ (same as for common variant GWAS, conventional genome-wide significance corrected for the number of traits)) in both the model with and without adjusting for the common variant PGS and effect sizes did not differ by more than 20% between these models, as this might otherwise indicate that rare variant findings cannot clearly be distinguished from common variant effects.

## Phenotype definition

To systematically test for phenotypic consequences of genes identified through rare variant analysis, we collated 626 disease entities following previous work[1] by aggregating information from self-report, HES, death certificates and primary care data (45% of the UKB population). Each disease entity had at least one significant common variant, and we used a similar analysis workflow using REGENIE as described for NMR measures but using logistic regression with saddle point approximation.

## Integration of OMIM

We downloaded the OMIM gene–disease list (9 November 2023) and kept 7,327 unique entries after filtering for gene entries with high confidence (level 3). We computed the enrichment of genes associated with any NMR measure from rare variant or gene burden analysis against a background of 19,989 protein coding genes using Fisher's exact test.

## Reporting summary

Further information on research design is available in the Nature Portfolio Reporting Summary linked to this article.

## Data availability

All individual-level data are publicly available to bona fide researchers via the UKB at https://www.ukbiobank.ac.uk/. Full summary statistics

for all analyses are publicly available through the NHGRI-EBI GWAS Catalogue (GWAS Catalog identifiers GCST90497044–GCST90501341; see GitHub repository).

## Code availability

Code for the main analyses is freely available via GitHub at https://github.com/comp-med/ukb-mgwas and permanently archived via Zenodo at https://doi.org/10.5281/zenodo.14716599 (ref. 92).

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

## Acknowledgements

We acknowledge the Scientific Computing of the IT Division at the Charité – Universitätsmedizin Berlin for providing computational resources that have contributed to the research results reported in this article (https://www.charite.de/en/research/research_support_services/research_infrastructure/science_it/#c30646061). We acknowledge Nightingale Health Plc for access to the UKB NMR biomarker data. We are deeply grateful to the participants, investigators and teams of the UKB and FinnGen studies. We thank B. Wild for assistance in data processing and helpful discussions. This work was supported by DZHK (German Centre for Cardiovascular Research) and BMBF (German Ministry of Education and Research) grants to C.L. and cofunded by a European Union grant to M.P. (ERC, GenDrug, 101116072) and supported by the Friede Springer Cardiovascular Prevention Center at Charité – Universitätsmedizin Berlin, Germany to A.W. M.M. is the British Heart Foundation Chair for Cardiovascular Proteomics (BHF Chair CH/16/3/32406, BHF Programme Grant RG/F/21/110053) and supported by the Imperial BHF Research Excellence Award (4) (RE/24/130023) and the VASCage Research Centre on Clinical Stroke Research, Austria. VASCage is a COMET Centre within the Competence Centers for Excellent Technologies (COMET) programme and funded by the Federal Ministry for Climate Action, Environment, Energy, Mobility, Innovation and Technology, the Federal Ministry of Labour and Economy, and the federal states of Tyrol, Salzburg, and Vienna. COMET is managed by the Austrian Research Promotion Agency (Österreichische Forschungsförderungsgesellschaft) FFG (project number 898252). Views and opinions expressed are, however, those of the authors only and do not necessarily reflect those of the European Union or the European Research Council. Neither the European Union nor the granting authority can be held responsible for them. The funders had no role in the study design, data collection and analysis, decision to publish or preparation of the manuscript.

## Author contributions

Conceptualization: M.Z., M.P. and C.L. Data curation/software: M.Z., C.B., S.Y., L.K. and M.P. Formal analysis: M.Z., C.B., S.Y., L.K., A.N., A.W. and M.P. Methodology: M.Z., C.B., S.Y., L.K., M.K., A.W., M.P. and C.L. Visualization: M.Z., C.B., L.K., A.W., M.P. and C.L. Funding acquisition: C.L. and M.P. Project administration: C.L. Supervision: M.P. and C.L. Writing—original draft: M.Z., C.B., S.Y., M.P. and C.L. Writing—review and editing: M.Z., C.B., S.Y., L.K., M.K., F.K., M.M., A.W., M.P. and C.L.

## Competing interests

The authors declare no competing interests.

## Additional information

**Extended data** is available for this paper at https://doi.org/10.1038/s41588-025-02355-3.

**Correspondence and requests for materials** should be addressed to Maik Pietzner or Claudia Langenberg.

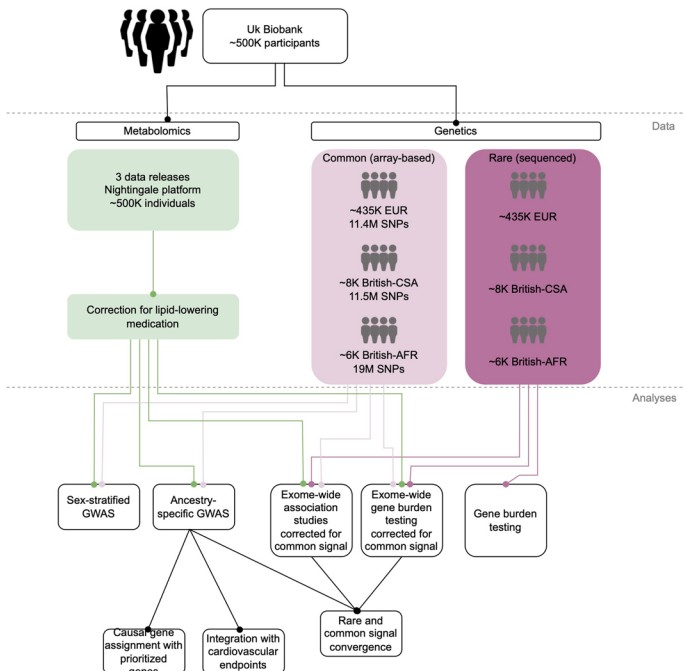

**Extended Data Fig. 1 | Graphical outline of the study design.** EUR, European ancestry; CSA, Central/South Asian ancestry; AFR, African ancestry.

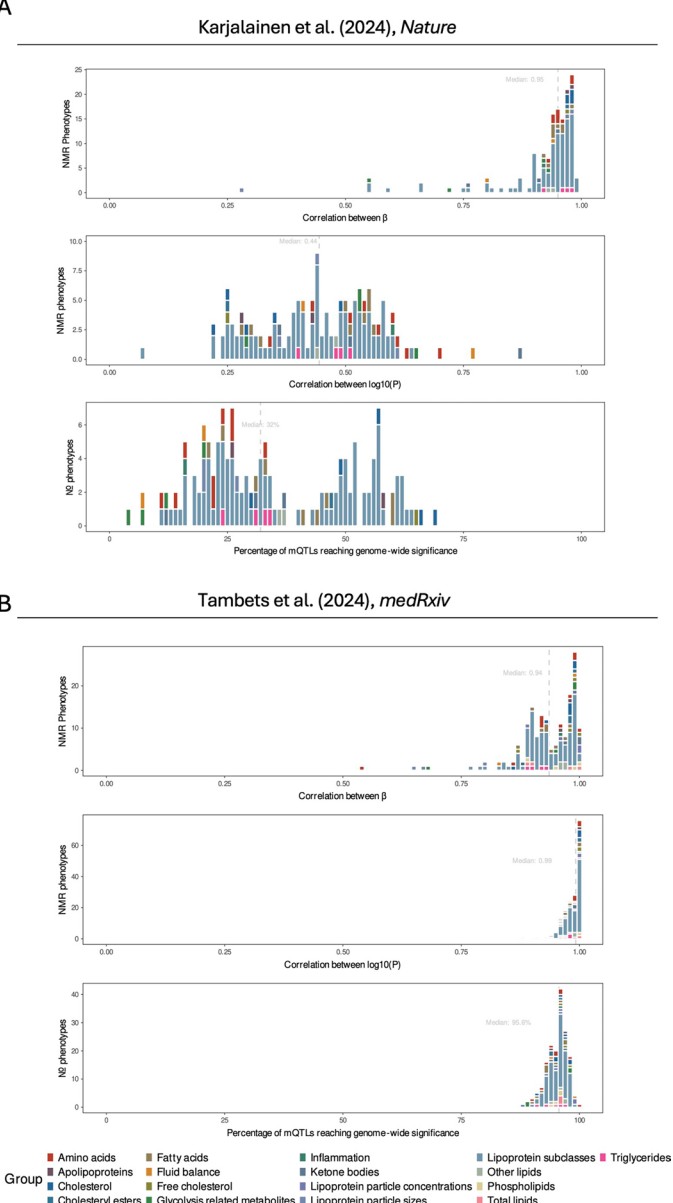

**Extended Data Fig. 2 | Independent replication of genetic signals.**
**a**, Replication of estimated genetic effects on circulating metabolites in Karjalainen et al.[3]. Bar plots represent the correlation of effect sizes (top), correlation between the *P*-value (middle), and the fraction of our sentinel variants that reached genome-wide significance in the replication study (bottom). **b**, Identical to **a** but using data from Tambets et al.[9]. For both comparisons, we only considered directly measured traits.

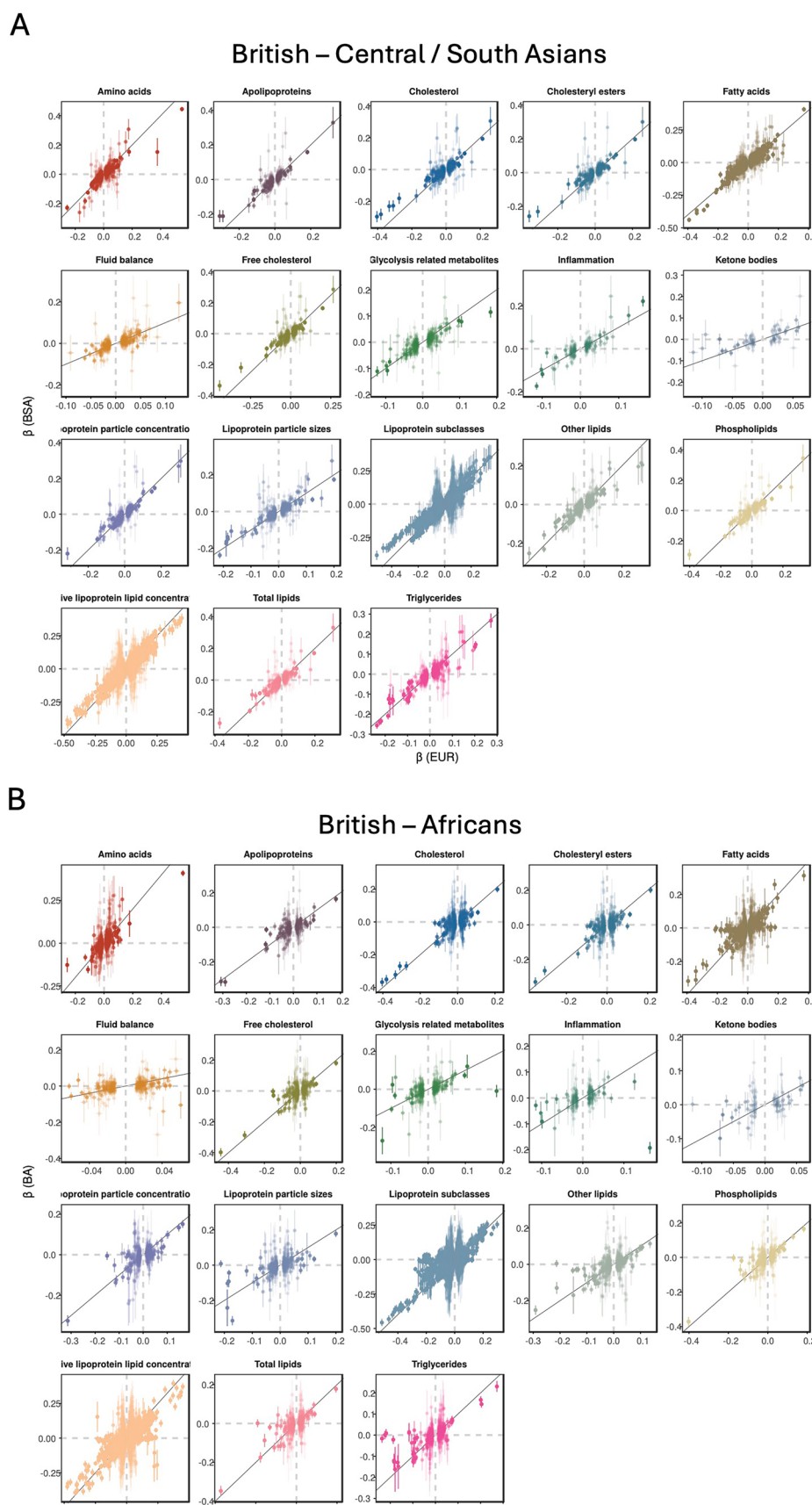

**Extended Data Fig. 3 | Cross-ancestry comparison of genetic effects.**
**a**, **b**, Cross-ancestry comparison of estimated genetic effects. Comparing estimates (points) obtained within UK Biobank participants of European ancestry (x-axis, n = 434,646) to those of British Central/South Asian ancestry (n = 8,796) (**a**) or British African ancestry (n = 6,573) (**b**). Bars denote standard errors of the estimates.

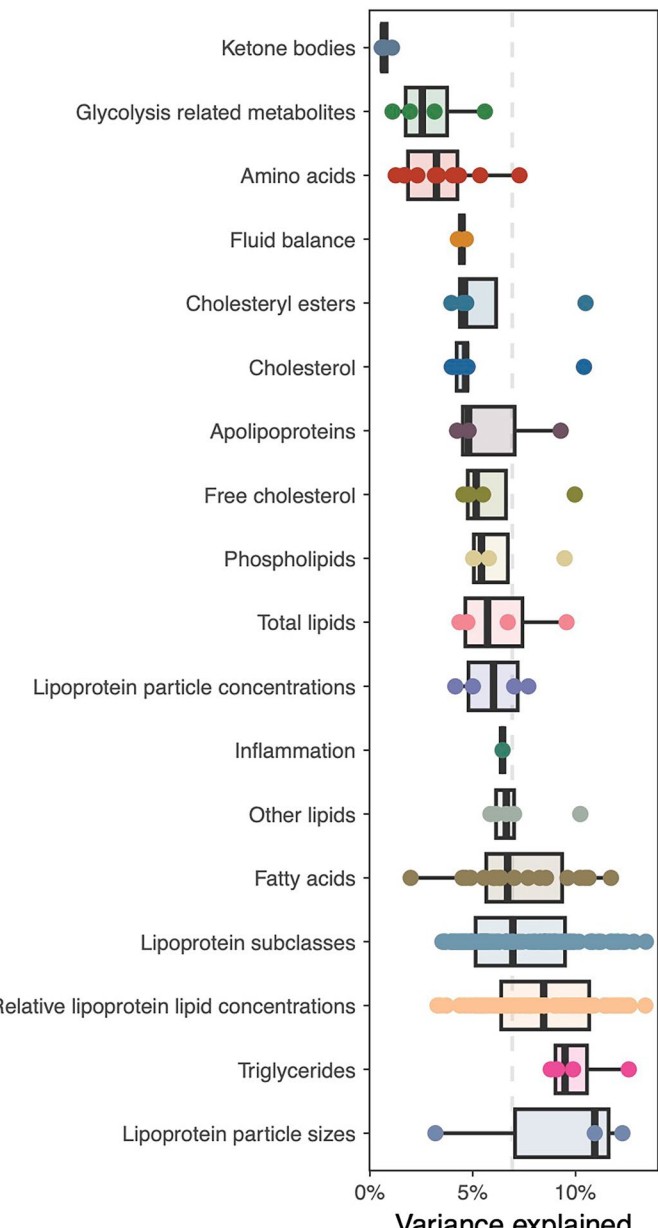

**Extended Data Fig. 4 | Plasma metabolome variance explained by genetics.** Variance explained by fine-mapped lead variants on metabolomic concentrations. Each dot represents a metabolite, colored for biochemical class. Boxplot center refers to the median, bounds are the upper and lower quartiles, and whiskers indicate 1.5× interquartile range.

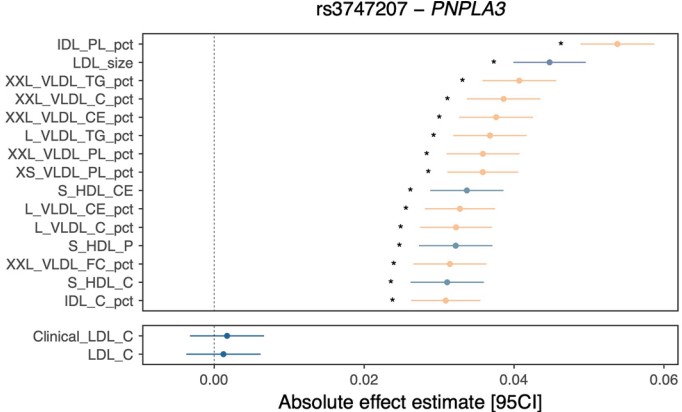

**Extended Data Fig. 5 | Association profile for *PNPLA3*.** Forest plot showing the strongest associated NMR traits for rs3747207, previously associated with LDL-cholesterol. Stars represent whether traits are significantly differently associated compared to LDL-cholesterol. Effect estimates (dots) and standard errors of the estimate (bars) are taken from the European ancestry-based GWAS (*n* = 434,646).

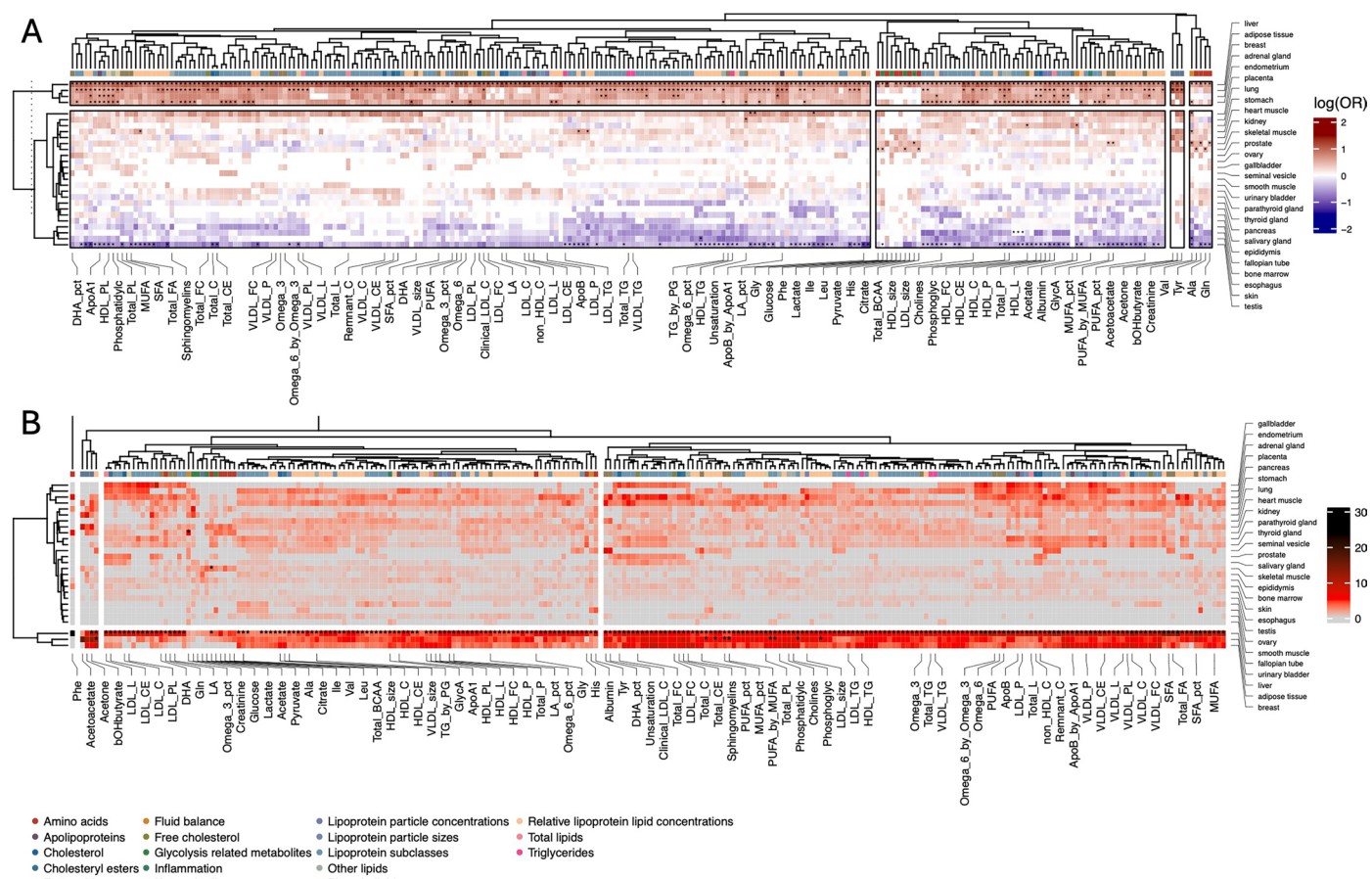

**Extended Data Fig. 6 | Effector gene tissue enrichment. a, b,** Odds ratios for enrichment of assigned effector genes (**a**) and genes proximal to fine-mapped lead variants (**b**) across tissue compartments. Columns represent each of the 249 metabolic traits, annotated for biochemical class. Rows and columns were clustered based on Euclidean distance. Odds ratios are derived from a two-sided Fisher's test.

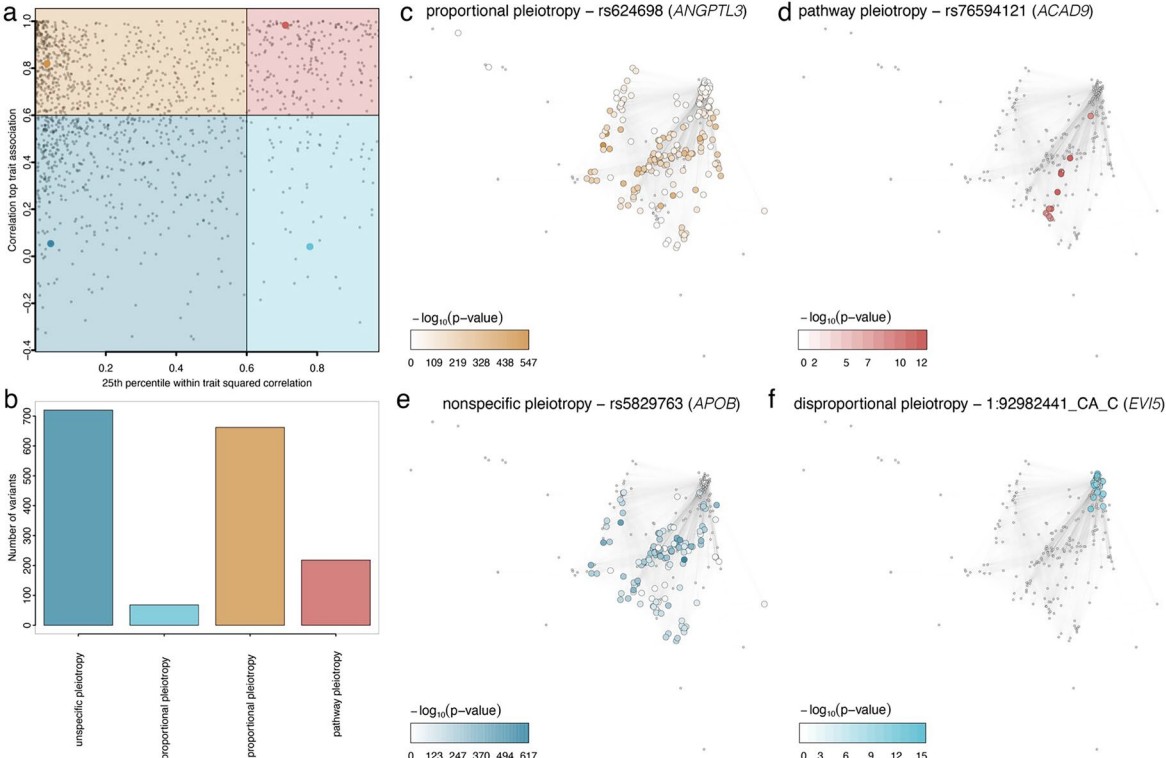

**Extended Data Fig. 7 | Different modes of metabolomic pleiotropy. a**, Scatterplot opposing mQTL characteristics. The *x*-axis denotes for each mQTL the 25th percentile of all possible correlations among associated NMR measures. The *y*-axis depicts the correlation between the strongest trait of interest and the association strength for all other traits. A value of one would indicate that all other associated NMR measures can be directly explained as function of correlation, whereas a value of zero would indicate independent effects of the mQTL on different measures. **b**, Bar plot showing number of variants for each mode of pleiotropy. **c-f**, Same Pearson correlation networks of NMR measures, clustering highly correlated traits by spatial proximity. Each node is colored according to the strength of associations ($-\log_{10}(P$-value)) with one of the four genetic variants indicated in the title of each plot. Variants were chosen to represent each of the four modes of pleiotropy.

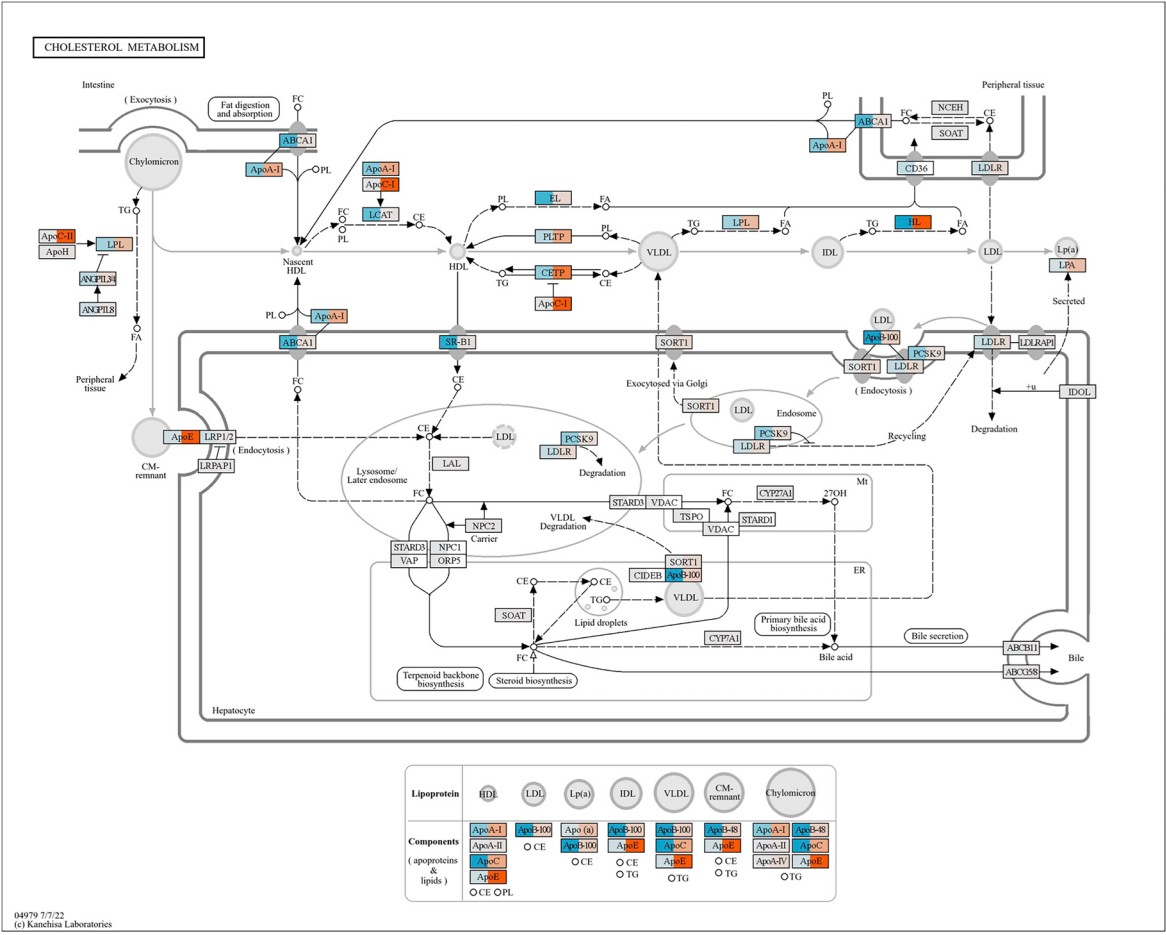

**Extended Data Fig. 8 | Rare and common variant convergence in metabolic genes.** Convergence of gene burden (blue) and common variant (orange) burden results for genes involved in cholesterol metabolism.

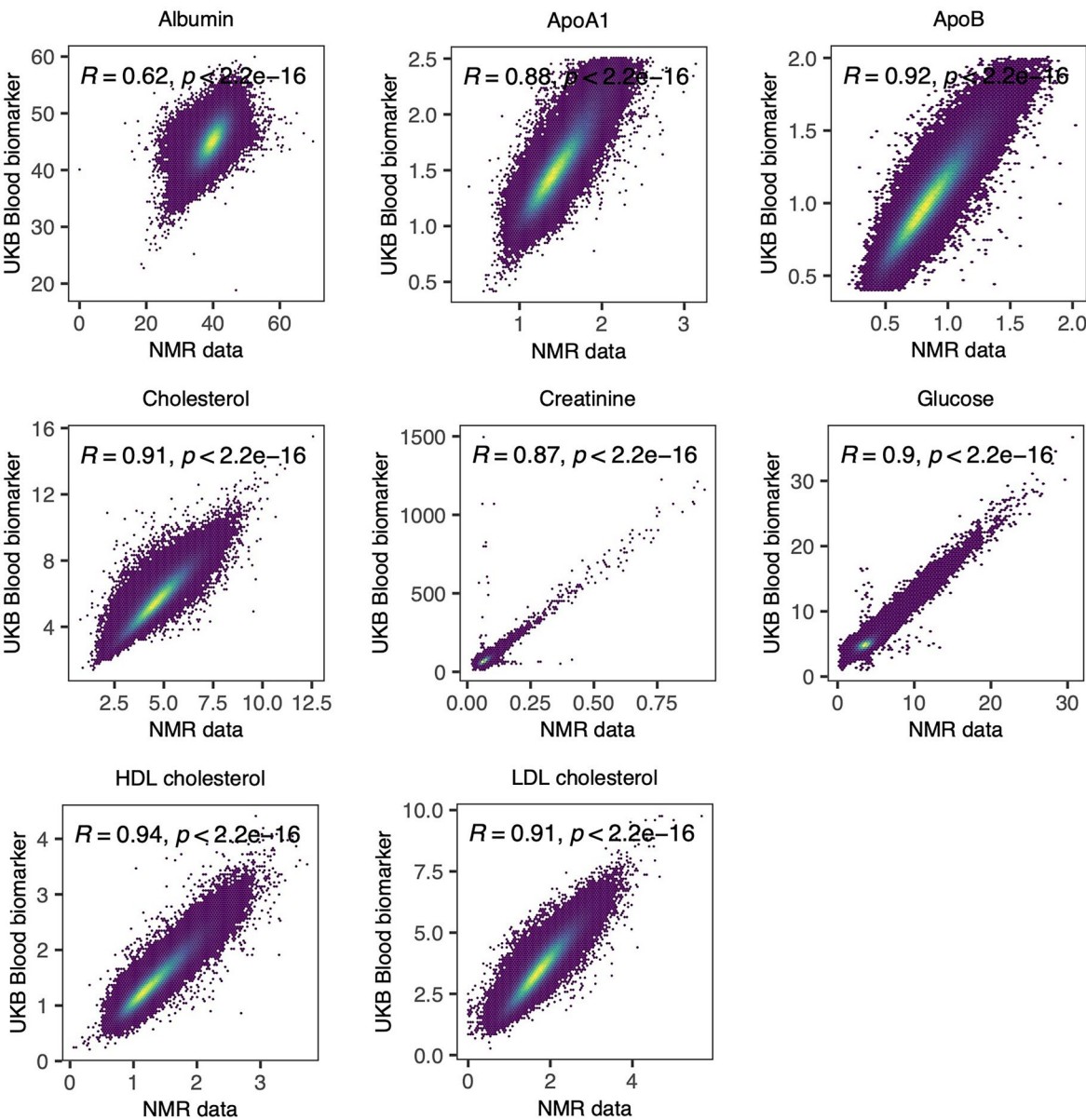

**Extended Data Fig. 9 | Comparison of NMR measures with blood biomarkers.** Comparison of eight metabolic traits measured on the NMR platform (*x* axis) overlapping with routine blood biomarkers previously measured in the same cohort (*y* axis).

# Reporting Summary

## Statistics

For all statistical analyses, confirm that the following items are present in the figure legend, table legend, main text, or Methods section.

| n/a | Confirmed | |
|---|---|---|
| ☐ | ☒ | The exact sample size (*n*) for each experimental group/condition, given as a discrete number and unit of measurement |
| ☐ | ☒ | A statement on whether measurements were taken from distinct samples or whether the same sample was measured repeatedly |
| ☐ | ☒ | The statistical test(s) used AND whether they are one- or two-sided *Only common tests should be described solely by name; describe more complex techniques in the Methods section.* |
| ☐ | ☒ | A description of all covariates tested |
| ☐ | ☒ | A description of any assumptions or corrections, such as tests of normality and adjustment for multiple comparisons |
| ☐ | ☒ | A full description of the statistical parameters including central tendency (e.g. means) or other basic estimates (e.g. regression coefficient) AND variation (e.g. standard deviation) or associated estimates of uncertainty (e.g. confidence intervals) |
| ☐ | ☒ | For null hypothesis testing, the test statistic (e.g. *F*, *t*, *r*) with confidence intervals, effect sizes, degrees of freedom and *P* value noted *Give P values as exact values whenever suitable.* |
| ☐ | ☒ | For Bayesian analysis, information on the choice of priors and Markov chain Monte Carlo settings |
| ☒ | ☐ | For hierarchical and complex designs, identification of the appropriate level for tests and full reporting of outcomes |
| ☐ | ☒ | Estimates of effect sizes (e.g. Cohen's *d*, Pearson's *r*), indicating how they were calculated |

*Our web collection on statistics for biologists contains articles on many of the points above.*

## Software and code

Policy information about availability of computer code

| Data collection | No software was used for data collection. |
|---|---|
| Data analysis | Data analyses were performed using REGENIE (v3.1.1), METAL (v2020-05-05), plink (v2), bcftools (v1.15.1), VEP (v106.1), REVEL, CADD (v1.6), LOFTEE, BEDtools (v2.3), LDStore v2. Further downstream analyses were implemented in R, using packages uknbnmr (v2.2), mice (v3.15), susieR (v0.12.35), igraph (v2.0.01), MungeSumStats (v1.13.2), coloc (v5.2.3), twoSampleMR (v0.5.1). Multi-ancestry finemapping was implemented in MultiSuSiE. Representative scripts used in analysis are freely available on Github (https://github.com/comp-med/ukb-mgwas) |

For manuscripts utilizing custom algorithms or software that are central to the research but not yet described in published literature, software must be made available to editors and reviewers. We strongly encourage code deposition in a community repository (e.g. GitHub). See the Nature Portfolio guidelines for submitting code & software for further information.

# Data

Policy information about availability of data

All manuscripts must include a data availability statement. This statement should provide the following information, where applicable:
- Accession codes, unique identifiers, or web links for publicly available datasets
- A description of any restrictions on data availability
- For clinical datasets or third party data, please ensure that the statement adheres to our policy

All individual-level data is publicly available to bona fide researchers from the UK Biobank (https://www.ukbiobank.ac.uk/). Full summary statistics for all analyses are publicly available through the NHGRI-EBI GWAS Catalogue (GWAS catalog identifiers GCST90497044 - GCST90501341, see Github repository)

# Research involving human participants, their data, or biological material

Policy information about studies with human participants or human data. See also policy information about sex, gender (identity/presentation), and sexual orientation and race, ethnicity and racism.

| | |
|---|---|
| Reporting on sex and gender | We defined 'female' and 'male' sex including participants where the recorded sex and sex chromosomes aligned (XX for females and XY for males). The recorded sex was self-reported, and it was not possible to distinguish sex from gender. We acknowledge the importance of distinguishing between sex and gender in research and that chromosomal make-up does not always align with self-identified gender. To assess whether our genetic analyses were driven by sex differences and whether our results were transferrable to both sexes, we performed sex-stratified GWAS within the largest ancestry (EUR). |
| Reporting on race, ethnicity, or other socially relevant groupings | We used previously published ancestral assignments by the pan-UKB consortium to assign individuals to ancestral groups, and made a further effort to .assign unclassified individuals to their respective ancestries based on a k-nearest neighbour approach using genetic principal components |
| Population characteristics | UK Biobank is a prospective cohort study from the UK that contains more than 500,000 volunteers between 40 and 69 years of age at inclusion. The cohort has been extensively described elsewhere (www.ukbiobank.ac.uk). Individuals were not directly selected for inclusion in the study on the basis of any disease or health parameter. The study consisted of 54.3% women and participants were on average 56.8 years old(s.d.:8.0). |
| Recruitment | All individuals between the age of 40-69 (men and women) who were registered with the National Health Service and living within a 25-mile radius from one of 22 recruitment centers spread across the United Kingdom were invited to participate in 2006-2010. Overall, about 9.2M individuals were invited to recruit around 0.5M individuals. |
| Ethics oversight | The UKBB was approved by the National Research Ethics Service Committee North West Multi-Centre Haydock. |

Note that full information on the approval of the study protocol must also be provided in the manuscript.

# Field-specific reporting

Please select the one below that is the best fit for your research. If you are not sure, read the appropriate sections before making your selection.

☒ Life sciences ☐ Behavioural & social sciences ☐ Ecological, evolutionary & environmental sciences

For a reference copy of the document with all sections, see nature.com/documents/nr-reporting-summary-flat.pdf

# Life sciences study design

All studies must disclose on these points even when the disclosure is negative.

| | |
|---|---|
| Sample size | We used the three major ancestral groups represented in the UK Biobank for our discovery analyses. The UKB (n≈500,000 individuals) is currently the largest available resource with linked genetic and metabolomics data. |
| Data exclusions | We excluded individuals where we did not have access to matching genotyping (array or sequencing-based) and metabolomics data. Furthermore, individual samples failing standard genotyping quality control or not assigned to one of the three major ancestral groups were excluded. These decisions were made before performing any statistical analysis. |
| Replication | We replicated our metabolome-wide genome-wide associations in the currently largest available meta-analysis of the same targeted metabolomics platform. Replication showed high concordance with the previously published studies. |
| Randomization | N/A - randomization occurred naturally as genetic variants were the exposure. |
| Blinding | N/A - genetic association testing does not require blinding as the effect of genetic variants on disease outcome is relative to the allele used as a reference. |

# Reporting for specific materials, systems and methods

We require information from authors about some types of materials, experimental systems and methods used in many studies. Here, indicate whether each material, system or method listed is relevant to your study. If you are not sure if a list item applies to your research, read the appropriate section before selecting a response.

## Materials & experimental systems

| n/a | Involved in the study |
|-----|-----------------------|
| ☒ | Antibodies |
| ☒ | Eukaryotic cell lines |
| ☒ | Palaeontology and archaeology |
| ☒ | Animals and other organisms |
| ☒ | Clinical data |
| ☒ | Dual use research of concern |
| ☒ | Plants |

## Methods

| n/a | Involved in the study |
|-----|-----------------------|
| ☒ | ChIP-seq |
| ☒ | Flow cytometry |
| ☒ | MRI-based neuroimaging |

## Plants

**Seed stocks**
*Report on the source of all seed stocks or other plant material used. If applicable, state the seed stock centre and catalogue number. If plant specimens were collected from the field, describe the collection location, date and sampling procedures.*

**Novel plant genotypes**
*Describe the methods by which all novel plant genotypes were produced. This includes those generated by transgenic approaches, gene editing, chemical/radiation-based mutagenesis and hybridization. For transgenic lines, describe the transformation method, the number of independent lines analyzed and the generation upon which experiments were performed. For gene-edited lines, describe the editor used, the endogenous sequence targeted for editing, the targeting guide RNA sequence (if applicable) and how the editor was applied.*

**Authentication**
*Describe any authentication procedures for each seed stock used or novel genotype generated. Describe any experiments used to assess the effect of a mutation and, where applicable, how potential secondary effects (e.g. second site T-DNA insertions, mosiacism, off-target gene editing) were examined.*

