## [Peer Review File · Nature Genetics]

A genetic map of human metabolism across the allele frequency spectrum

Corresponding Author: Professor Claudia Langenberg

Version 0:

Decision Letter:

8th Nov 2024

Dear Claudia,

Thank you for submitting your manuscript entitled "A genetic map of human metabolism across the allele frequency spectrum". As noted, before we can send your study for review, we would like these reporting checklists to be ready.

We want to ensure that the methods and statistics reporting in our papers are of the highest quality. To that end, we ask authors to fill out a reporting summary that collects information on experimental design and reagents. The Reporting Summary and Editorial Policy Checklist can be found using the links below:

-- Reporting summary: <https://www.nature.com/documents/nr-reporting-summary.pdf>

-- Editorial policy checklist: <https://www.nature.com/documents/nr-editorial-policy-checklist.pdf>

Please be aware of our guidelines on digital image standards.

Please complete the forms and return them as soon as possible. Please note that the forms are dynamic 'smart pdfs' and must therefore be downloaded and completed in Adobe Reader. We will then flatten it for ease of use by the reviewers. If you would like to reference the guidance text as you complete the template, please access these flattened versions at <http://www.nature.com/authors/policies/availability.html>.

Note that you are not required to revise your paper to include the information provided in the reporting summary. However, all points on the policy checklist must be addressed; please send me a new version of the manuscript with your completed checklist if needed.

If you anticipate a delay of more than four weeks, please let us know. We will be happy to consider your revision so long as nothing similar has been accepted for publication at Nature Genetics or published elsewhere. Should your manuscript be substantially delayed without notifying us in advance and your article is eventually published, the received date may be that of the revised, not the original, version.

If you are not interested in submitting a suitably revised manuscript in the future please let me know immediately so we can close your file. If you have any questions, please contact me.

Please use the link below when you are prepared to resubmit.

Link Redacted

Thank you for your interest in Nature Genetics.

Sincerely,

Michael Fletcher, PhD
Senior Editor, Nature Genetics
ORCID: 0000-0003-1589-7087

Version 1:

Decision Letter:

10th Dec 2024

Dear Claudia,

Your Article, "A genetic map of human metabolism across the allele frequency spectrum" has now been seen by 3 referees. You will see from their comments below that while they find your work of interest, some important points are raised. We are interested in the possibility of publishing your study in Nature Genetics, but would like to consider your response to these concerns in the form of a revised manuscript before we make a final decision on publication.

In brief, the reviewers sound overall enthusiastic for publication of your work at the journal, although it is also clear that some improvement is yet required before a final decision.

Referee #1 provides a succinct report that is largely positive; they have a suggestion for a few further interesting analyses (e.g. coloc with pQTLs) and also requests sharing of a comprehensive set of summary statistics. Reviewer #2 sounds less supportive, calling into question some of the highlighted examples, and also suggesting that the novelty over other recent studies is unclear; they provide some guidance to improve these aspects. Reviewer #3 comes across similarly to #1, again suggesting summary statistics must be comprehensively shared. Their other requests are largely presentational, rather than requesting further analysis, although we note they also suggest some of the described examples are unconvincing.

In our reading, there is a clear path to publication, and none of the requests seem unduly onerous or impractical. We think that Referee #2 is angling for you and your co-authors to more clearly demonstrate the novelty of your work, although we find it hard to judge whether the various requests for additional analysis will lead to a satisfactory improvement for them; nonetheless, given the positivity of the other referees this is unlikely to be an obstacle to a positive decision. We would also emphasise important overlapping requests that should be comprehensively addressed by a revision, most notably the summary statistic sharing (and ideally to the level of granularity and web-accessibility requested), as well as the biological plausibility of specific examples described in detail (e.g. VEGFA).

To guide the scope of the revisions, the editors discuss the referee reports in detail within the team, including with the chief editor, with a view to identifying key priorities that should be addressed in revision and sometimes overruling referee requests that are deemed beyond the scope of the current study. We hope that you will find the prioritized set of referee points to be useful when revising your study. Please do not hesitate to get in touch if you would like to discuss these issues further.

We therefore invite you to revise your manuscript taking into account all reviewer and editor comments. Please highlight all changes in the manuscript text file. At this stage we will need you to upload a copy of the manuscript in MS Word .docx or similar editable format.

*2) If you have not done so already please begin to revise your manuscript so that it conforms to our Article format instructions, available

http://www.nature.com/ng/authors/article_types/index.html

*3) Include a revised version of any required Reporting Summary: <https://www.nature.com/documents/nr-reporting-summary.pdf>

It will be available to referees (and, potentially, statisticians) to aid in their evaluation if the manuscript goes back for peer

review.

Link Redacted

Sincerely,

Michael Fletcher, PhD
Senior Editor, Nature Genetics
ORCID: 0000-0003-1589-7087

Referee expertise: human genetics; metabolomics

Reviewers' Comments:

Reviewer #1 (Remarks to the Author):

The author present association of common and rare sequence variants with 249 metabolic phenotypes among half a million UK Biobank participants of 3 ancestries.

This manuscript and this resource will be very valuable for a wide range of researchers. The results contain a large number of original and novel results. The methodology appears robust.

1/ The authors have a unique opportunity to study systematically mQTL in each of the three ancestries. Can the author quantify how many association are with variants almost exclusively represented by individuals of non-European ancestries. The example of rs3211938 within CD36 associating with Omega 3 is interesting but how often such example can be found. Here it is not about heterogeneity in effect but in frequencies.

2/ In figure 2 C about refinement, can the authors add a locus plot of association not just of probabilities

3/ The authors have previously reported of pQTL in large sets. Have they attempted to assess colocalization of pQTL previously reported and mQTL detected in that study.

4/ The authors mentioned that the rare variants they study are based on the exome sequencing. Is their aa specific rationale not to use data from whole genome sequencing

5/ I suggest that the authors have the number of tested metabolic phenotypes tested in the abstract

6/ In Figure 1 a the dots are colored, the label for the colors should be presented in the same order as presented in 1a, and not alphabetically

There is a total of 18 colors and it is easy to get confused if not ordered in the same way in graph and label

7/ Do the authors present in the table the result per ancestry in addition to trans ancestry. This would be very helpful. Similarly when releasing GWAS summary statistics this would be of high value to have the trans ancestry as well as the 3 ancestry specific

8/ If the authors have some plan to host the results in a web browser, I would strongly suggest to also present results per ancestry

Reviewer #2 (Remarks to the Author):

Zoodsma et al have performed a genome-wide association study of NMR-measured small molecules and lipid and lipoprotein measures. The authors used large-scale genetic and metabolomics data from the UK Biobank. They identified 753 genomic regions associated with metabolites. The majority of the associations were novel (at least according to them), and several examples of interesting novel loci are presented. They analyzed patterns of pleiotropy with a newly developed framework and identified potential effector genes using machine learning. They suggest that their results can guide, e.g., the selection of drug targets, and indicate VEGFA as a possible modulator of coronary artery disease.

The manuscript represents an important effort, is mostly well written (although maybe too complicated and detailed at some paragraphs) and provides novel aspects into disease biology. Most statistical analyses seem to be robustly conducted. I particularly appreciate that the authors give several examples of novel loci and discuss their biological importance. However, recent NMR GWA studies have had larger sample sizes. There are also some additional concerns. Please see my comments below:

Major comments:

1. The authors write: "Here we present the most comprehensive study of human metabolism to date..." However, a larger GWAS of NMR metabolites (using Estonian and UK Biobank data) was recently published as a preprint (Tambets et al. 2024; <https://www.medrxiv.org/content/10.1101/2024.10.15.24315557v2.full>), and the GWAS summary statistics of that study are publicly available. There is also another recent preprint describing GWAS of NMR metabolites in UKBB (van der Meer et al. 2024; <https://www.medrxiv.org/content/10.1101/2024.07.30.24311254v1>). In the light of the study by Tambets et al, the authors should probably modify the text (e.g., statements about the number of novel loci) throughout the manuscript and focus the contents of the manuscript on those aspects that were not covered in the preprint by Tambets et al (and van der Meer et al). Having said that, I would like to point out that the size of the study population should not be the only thing that matters. In particular, I wish to point out that compared to the preprints by Tambets et al and van der Meer et al, this study has at least one very important strength: Zoodsma et al give examples of associated loci throughout the manuscript and discuss the biological relevance of their findings. This is exactly what should be done, instead of just listing statistical associations (like many studies in the field do), and the authors should be acknowledged for this.
2. Replication was assessed by comparing to the previous largest NMR GWA study (Karjalainen et al 2024, Nature 628). The authors state that only 24.8% of their associations reached GW significance in the previous study. What about replication considering higher p-value thresholds: $p < 10^{-5}$ (or even $p < 0.05$)? How many of the associations are replicated when comparing with the European results? What could be the reasons for non-replication, or is it just due to lower power in the previous study? Importantly, it should be discussed what the overall novelty of this study is compared to the study by Karjalainen et al.
3. Some of the Mendelian randomization analyses are likely to be confounded by pleiotropy (e.g. Fig 6A). The authors have included SNPs from all associated loci in the MR analyses. However, most of the SNPs associated with the lipid and lipoprotein traits are very pleiotropic, and therefore the results of the MR analyses are not reliable. The authors write that they have also used lead credible sets restricted for molecular pleiotropy, but it is not clear how they accounted for pleiotropy, and in which MR analyses these were used.
4. In my opinion, too much focus is given to the rare variant findings. I would suggest shortening those parts.
5. The paragraph "Risk mitigation of atherosclerotic cardiovascular disease beyond LDL-cholesterol" is hard to follow. Discussion and results related to VEGFA as a potential target of medication appear to be quite speculative (e.g., Fig 6), based on the current findings, if the authors are suggesting this mainly based on their suggestion of HDL particle size being causally related to CAD risk (?). Majority of research is not indicating HDL-related pathways causal for CAD, so such finding from MR analysis should be interpreted with caution.

Minor comments:

1. Abstract: Please state the number of metabolites analyzed. Please give some examples of novel loci detected.
2. Fig 1A is very hard to read. Please consider alternative ways to summarize your findings. Maybe include Fig1B-D in Supplement.
3. Fig. 2 could be in the supplement.
4. Suppl Table 1: please organize the metabolite traits in a logical way.
5. Could the authors comment on how their causal/effector gene annotations (Suppl Tables 2, 4 and/or 6) correspond to those indicated in the previous largest NMR GWA study (Karjalainen et al 2024, Nature 628). It would be good to include at least some examples/comparisons.
6. The paragraph describing the results of pleiotropy analyses would benefit from shortening. The same applies to the paragraph describing the convergence of common and rare variation.
7. Fig 4 is difficult to follow. For example, what does a single dot in the scatter plots a-d represent? In the upper part of Fig 4e, are only selected locus names shown? Fig. 4f and 4g would better be put into supplement. Maybe make Fig 4e clearer (for example, make the lines thicker if possible) and move other parts of the figure to supplement?
8. Metabolite measurements in Fig 5b should be organized in a logical way. Are the metabolite ratios/percentages the most relevant traits to show? Maybe concentrate on the other measurements. Maybe put Fig 5e and 5f into supplement?
9. Fig 6 – what does it mean to be a "not-LDL associated locus"? How is this defined? Maybe by comparison to largest available LDL cholesterol GWAS? Fig 6d does not give any information as such. Maybe present in a different way, or pick examples of most relevant associations?

10. Could locuszoom plots be shown for exemplary loci with interesting colocalizing metabolite and trait associations?
11. Please discuss the pros and cons of using a single cohort in this kind of GWAS.

Reviewer #3 (Remarks to the Author):

This manuscript reports on the genetic architecture of metabolomic traits as quantified by H1-NMR (Nightingale platform) among approximately 450,000 participants of the UK Biobank (UKB) of British White European (EUR), British African (BA) and British Central South Asian (BSA) ancestries. The authors provide a comprehensive catalogue of approximately 30,000 locus-metabolite signals mapping to 753 regions, with effector gene assignment, characterization of pleiotropy, establishment of allelic series, and assessment of how these findings relate to human disease. The detected common variant associations show largely consistent effect sizes across ancestries and across sexes.

This group of authors is very experienced with genome-wide screens of intermediate molecular traits such as metabolites or proteins. This is also evidenced by their group obtaining early access to the UKB Nightingale measurements. The analyses are state-of-the-art, and the authors did an excellent job on the thorough exploration of the metabolome, phenome and genome. The manuscript is very well written, with nice graphics. I do not have any substantial general concerns about the presented findings, although it is somewhat of a pity that - despite the abundance of significant results - many findings are expected and confirmatory and few insights directly translate into altered disease risk, or - if they are unexpected and novel - it is difficult to know if they really represent new causal relationships (see below). However, this is not the authors' fault, and the presented findings represent a robust and very valuable resource for the scientific community.

The authors are asked to address the comments below, with major points related to the choice of showcases and the amount of speculation in the results and discussion section.

Major:

1. Figure 2c: although there are many fewer variants in the credible set (CS) in multi-ancestry finemapping, the PIP of the best variant in EUR is not much lower than the PIP in multi-ancestry finemapping. Importantly, it is also just as centered on the PTRF gene, which also happens to be the closest gene anyway. Could the authors highlight another example that better illustrates improved resolution of the most likely causal gene owed to multi-ancestry finemapping?
2. Machine-learning effector gene assignment: although the authors report statistics in favor of their RF classifiers, the proportion of loci in which a significant gene-burden finding supports the assigned prioritized gene is disappointingly low (not even a third of them is among the top three ML-based assigned effector genes). How do the authors explain this? Given the evidence through human support via pLoF variants, wouldn't it be more straightforward to assign the gene implicated by the burden test as the causal gene rather than relying on an algorithm that includes evidence from pervasive eQTLs etc.? How do the enrichment statistics reported throughout the paper compare when these analyses are repeated with the genes supported by significant gene burden tests prioritized as effector genes?
3. Lines 407-410: the authors speculate that the association between pLoF variants in SLC30A10 and lipoprotein particle composition may be mediated by manganese (Mn). Can they investigate whether the same variants driving the association with NMR particles are also related to blood Mn levels, using either GWAS data such as PMID: 38594418, or biobank sequencing data in case Mn measurements are available? If unavailable, the authors can test whether mutations in SLC30A10 known to cause the monogenic condition *Hyper manganeseemia With Dystonia 1* are present (at least in the heterozygous state) in the UKB and whether carrier have changes in the implicated metabolites in the expected direction.
4. Lines 494-508. I cannot follow the authors' logic around the SLC22A2 and creatinine/CKD showcase. The authors found that pLoF variants in the gene known to encode a creatinine transport protein, SLC22A2, are associated with creatinine levels and with chronic kidney disease, which is commonly defined based on the glomerular filtration rate estimated from creatinine. The variants are not associated with an alternative and superior marker of kidney function, serum cystatin C. The most likely explanation then is that the detected variants lead to differences in creatinine metabolism (and kidney disease defined from creatinine), but not truly to kidney dysfunction or kidney disease. If these variants were associated with kidney disease, then there would be an association with cystatin C levels. Cystatin C is available in the UKB, and the authors can and should test the associations of their rare variant masks with cystatin C levels. If it turns out that this finding mainly represents an effect on creatinine metabolism, then it is not the best showcase for the abstract, a main figure, and a full paragraph. Others have anyway reported rare LoF variants in this gene in association with creatinine-based kidney function in the UKB before (PMID: 36890159).
5. Lines 530-552: the authors speculate on the potential of activating VEGFA to influence properties of HDL particles as a potential mechanism to reduce CAD risk. However, the manuscript cannot establish that these properties actually mediate CAD risk and are not only proxies for unmeasured molecules, nor that one of the other lipoprotein properties that colocalize with CAD risk at VEGFA such as LDL cholesterol and triglyceride properties (Suppl Table 13) are the ones that are important. Further, MR results in ST 12 suggest that HDL cholesterol levels show just as strong potentially causal associations with CAD risk as do other HDL particle properties. Moreover, VEGFA is a potent angiogenesis factor that is activated in many cancers, raising questions if it is a worthwhile target to try to activate. I am not convinced that this is the strongest example to highlight in the abstract and a full paragraph of the results and in a main figure and suggest replacing it or toning it down.
6. Lines 589-595: the last two sentences are redundant, or something is missing. More importantly, if the implicated metabolite only proxies a disease state such as chronic inflammation, then how much can we rely on MR results that implicate potentially "causal" associations? If in addition we do not know how this metabolite relates to type 2 diabetes, again this is too much speculation for a results section of a manuscript.
7. Discussion, lines 628-631: true, but the authors do not know whether the NMR-based properties that they study are actually the ones on which the genes primarily act. They can now be measured and hence used for genetic association studies, but experimental proofs they represent the biologically important properties is mostly missing. I suggest removing

this sentence.

8. Data availability: GWAS summary statistics and the results from genome-wide burden tests need to be shared with the scientific community, i.e., made available for download or submitted to the GWAS catalogue or similar. It is not enough to say that they will be made available upon publication, it is important to know how they will be shared. The authors can set up temporary links or protected access so that reviewers can see how they are planning to share these results.

9. Methods: lines 819 ff. Is it correct that the joint statistics were computed in a different set of individuals (unrelated EUR) compared to the marginal Susie statistics coming from the overall sample? If so, then it does not seem to correct to compare potential attenuation of effect sizes, since the statistics stem from a partially non-overlapping sample.

Minor:

1. The authors compare effect sizes between EUR and BA or BSA, respectively (Fig 1c,d). Can they please also compare the results from the trans-ethnic meta-analysis and the EUR subset, for all trans-ethnic sentinel variants? In panel 1c, what is the subgroup of dots close to the horizontal line – do they all come from one or few very pleiotropic loci, giving rise to many dots that do not align?

2. Lines 171-173: how do the authors explain this substantial increase in the number of high-confidence variants upon multi-ancestry finemapping? Were these variants members of the credible sets in EUR already (should be, since these are larger) and only had lower PIPs?

3. Lines 239-241: please include the discovery trait in the GLGC in the text. Presumably it is LDL cholesterol levels, but it is not clear from the text. If so, how do the authors explain that have completely insignificant p-values in their analysis: is it only the smaller sample size? Please include the betas, not just the p-values.

4. Methods: Finemapping: if I did not miss it, please add which reference panels were used for finemapping in BA and BSA?

5. Lines 420-23: a potential role of NDRG2 in histidine metabolism is highly speculative at this point, without any experimental evidence, even in the literature. It would be better to replicate this association when highlighting it in the manuscript. It can happen that metabolites are mis-annotated from a specific method such as NMR. If other associations in this work support the correct annotation of histidine (e.g., with variants in HAL, I didn't check), then I would mention this in this context.

6. Figure 5d is a great panel. The authors should be commended on integrating common and rare variant evidence.

Version 2:

Decision Letter:

25th Feb 2025

Dear Claudia,

Your Article, "A genetic map of human metabolism across the allele frequency spectrum" has now been seen by the original 3 referees. You will see from their comments below that while they continue to find your work of interest, there are still some important points raised. We are interested in the possibility of publishing your study in Nature Genetics, but would like to consider your response to these concerns in the form of a revised manuscript before we make a final decision on publication.

Briefly, Referees #1 and #2 are satisfied and have no remaining comments. Reviewer #3, conversely, appreciates the improvement but thinks there are still a few further analyses and changes required before they will support publication.

In our reading these seem relatively slight requests and we thus encourage you and your co-authors to respond to them in full.

To guide the scope of the revisions, the editors discuss the referee reports in detail within the team, including with the chief editor, with a view to identifying key priorities that should be addressed in revision and sometimes overruling referee requests that are deemed beyond the scope of the current study. We hope that you will find the prioritized set of referee points to be useful when revising your study. Please do not hesitate to get in touch if you would like to discuss these issues further.

We therefore invite you to revise your manuscript taking into account all reviewer and editor comments. Please highlight all changes in the manuscript text file. At this stage we will need you to upload a copy of the manuscript in MS Word .docx or similar editable format.

*2) If you have not done so already please begin to revise your manuscript so that it conforms to our Article format

instructions, available

[here](http://www.nature.com/ng/authors/article_types/index.html).

*3) Include a revised version of any required Reporting Summary: <https://www.nature.com/documents/nr-reporting-summary.pdf>

Please be aware of our [guidelines](https://www.nature.com/nature-research/editorial-policies/image-integrity) on digital image standards.

EXTENDED DATA FIGURES

Link Redacted

We hope to receive your revised manuscript within four to eight weeks. If you cannot send it within this time, please let us know.

Sincerely,

Michael Fletcher, PhD
Senior Editor, Nature Genetics
ORCID: 0000-0003-1589-7087

Reviewers' Comments:

Reviewer #1 (Remarks to the Author):

The authors have answered all my points.
The manuscript is improved.

Reviewer #2 (Remarks to the Author):

The manuscript has clearly been improved after revision.

Reviewer #3 (Remarks to the Author):

The authors have thoroughly revised their manuscript and addressed many of my initial comments. Especially comments related to biological plausibility of speculative nature or showcases of loci not supported by sufficient evidence have been

revised (mostly be removal). Important additions are a comparison of findings to those from other studies, and a significantly revised and (in my opinion) improved workflow to assign effector genes by considering both common and rare variant support for a given gene, and the sharing of all summary statistics supporting the results.

However, several aspects require further attention and potentially modification:

Major:

- Newly added showcase on prioritization of PEPD as the effector gene (p.9, line 268 ff): a look at Suppl Table 6 shows that for the locus tagged by rs62102718, there was only one candidate gene (col H) in the region anyway, whereas metabolite associations tagged by other SNPs in this locus also list CEBPA as a potential candidate gene. If there is only one candidate and the lead variant is sitting in an intron of the assigned gene, then is this the best showcase to support the value of this prioritization workflow?
- New JAK2 example (p.14): the authors could improve their presentation of this finding by searching for more literature supporting a role of JAK2 in lipid and hepatic metabolism. A quick search shows that such literature may be published, in addition to the mentions CHIP connection (e.g., <https://pubmed.ncbi.nlm.nih.gov/23782652/>).
- New/revised Figure 5: this is interesting. For the regression estimates presented in panels e and f, more information would help. For instance, was there any overlap in carriers of the different variants? If so, then conditionally independent effect estimates should be presented, which may turn out important especially when effect directions can differ across variants.
- Limitations, p.23: it is an important limitation that NMR measurements of apolipoproteins can be affected in the presence of rare genetic variants. Given the focus on rare variants in the APOA1 gene in the main MS, in particular in Figure 5, their association with apolipoprotein A1 levels and other NMR metabolites, and the availability of apolipoprotein A measurements by classical blood biochemistry in the UKB, the authors should support these findings by using the blood biochemistry measurements in a sensitivity analysis. Please also show the correlation between "standard" and NMR apoA1 levels.

Minor:

- Figure 2 has been removed from the main MS, subsequent figures need to be renumbered.
- Please include in the main text that all regions found in non-EUR groups, other than the one at CD36 identified exclusively identified in BA participants, were also observed in the EUR participants.

Version 3:

Decision Letter:

Our ref: NG-A67108R2

25th June 2025

Dear Claudia,

Thank you for submitting your revised manuscript "A genetic map of human metabolism across the allele frequency spectrum" (NG-A67108R2). In light of the changes incorporated in response to Reviewer #3, we will be happy in principle to publish your study in Nature Genetics as an Article pending final revisions to comply with our editorial and formatting guidelines.

We are now performing detailed checks on your paper, and we will send you a checklist detailing our editorial and formatting requirements soon. Please do not upload the final materials or make any revisions until you receive this additional information from us.

Thank you again for your interest in Nature Genetics. Please do not hesitate to contact me if you have any questions.

Sincerely,
Kyle

Kyle Vogan, PhD
Senior Editor
Nature Genetics
<https://orcid.org/0000-0001-9565-9665>

Reviewers' comments

We are very grateful for the opportunity to revise the work and would like to thank the reviewers for their time and considerations that have substantially improved the quality of the paper. We provide here point-by-point answers addressing each suggestion. Responses and corresponding changes in the revised manuscript are marked in blue.

Reviewer #1 (Remarks to the Author): The author present association of common and rare sequence variants with 249 metabolic phenotypes among half a million UK Biobank participants of 3 ancestries. This manuscript and this resource will be very valuable for a wide range of researchers. The results contain a large number of original and novel results. The methodology appears robust.

1/ The authors have a unique opportunity to study systematically mQTL in each of the three ancestries. Can the author quantify how many associations are with variants almost exclusively represented by individuals of non-European ancestries. The example of rs3211938 within CD36 associating with Omega 3 is interesting but how often such example can be found. Here it is not about heterogeneity in effect but in frequencies.

R1 response 1 We identify a genetic region (the CD36 region mentioned by the reviewer) associated with 22 metabolic traits in British African participants but not seen in the largest group of European participants when comparing associations across the three largest ancestral groups represented in the UKB. All other regions associated with metabolites in the two smaller ancestral groups (British-Central/South Asian or British-African) are also seen in European participants (conventional genome-wide significance, $p < 5 \times 10^{-8}$).

The sentinel variants we identified in the smaller ancestral groups are frequently occurring signals across the populations (Figure 1), and we currently lack the statistical power to identify population-specific mQTLs (e.g. those exclusively seen in one population). More diverse biobanks will increase genuine discovery in underrepresented ancestral groups.

Figure 1: Effect allele frequencies of sentinel variants seen in British African (left) and British Central/South Asian individuals (right) compared to the European allele frequencies.

2/In figure 2 C about refinement, can the authors add a locus plot of association not just of probabilities.

R1 response 2 Following the reviewer's suggestion, we have added the marginal p-values from our genome-wide association studies to Fig. 2C (now Supplementary Figure 6). Given the overall length of the manuscript, added text from our revision, and request to shorten the text overall, we propose to follow the suggestion from R2 (minor comment 3) and move the revised figure to the supplemental material.

3/The authors have previously reported of pQTL in large sets. Have they attempted to assess colocalization of pQTL previously reported and mQTL detected in that study.

R1 response 3 *We followed this suggestion and have now included robust evidence for >4000 mQTL:pQTL colocalising signals (posterior probability for a shared signal >80%). These signals encompass 105 independent genetic regions associated with 249 metabolites colocalising with 145 unique protein targets. This provided independent support for effector gene assignment at 81 out of 143 loci. We have included a reference to these new results on p9, line 264.*

4/ The authors mentioned that the rare variants they study are based on the exome sequencing. Is their aa specific rationale not to use data from whole genome sequencing

R1 response 4 *We agree with the reviewer that use of whole genome sequencing (WGS) data would have been scientifically advantageous. Since UKB WGS data is currently still limited to pVCFs files, the quality control and formatting steps necessary prior to analyses would have been prohibitively expensive on the DNA Nexus platform for our team (estimated costs >100k GBP). Once these data become available in a research-ready format, this will become an option in the future. We note that for known variants, it has been shown that WGS data is not sufficiently superior to deep imputation based genotyped variants combined with exome sequencing (Gaynor et al, Nat Gen 2024).*

5/I suggest that the authors have the number of tested metabolic phenotypes tested in the abstract

R1 response 5 *We added the number of tested metabolites to the revised version of the abstract.*

6/ In Figure 1a. the dots are colored, the label for the colors should be presented in the same order as presented in 1a, and not alphabetically. There is a total of 18 colors and it is easy to get confused if not ordered in the same way in graph and label

R1 response 6 *We completely agree with the reviewer and accordingly adjusted the label for the colouring in all figures to follow the order in Fig. 1a.*

7/ Do the authors present in the table the result per ancestry in addition to trans ancestry. This would be very helpful. Similarly, when releasing GWAS summary statistics this would be of high value to have the trans ancestry as well as the 3 ancestry specific

R1 response 7 *Following the reviewer's suggestion, we now provide ancestry-specific results in Supplementary Table 2. We have also made available full summary statistics, including ancestry-specific analyses, in the GWAS catalogue.*

8/ If the authors have some plan to host the results in a web browser, I would strongly suggest to also present results per ancestry

R1 response 8 *To enable the immediate and widespread use of the results by the community, we have now uploaded all summary statistics, including those run in different ancestral strata to the GWAS catalog.*

Reviewer #2 (Remarks to the Author): Zoodsma et al have performed a genome-wide association study of NMR-measured small molecules and lipid and lipoprotein measures. The authors used large-scale genetic and metabolomics data from the UK Biobank. They identified 753 genomic regions associated with metabolites. The majority of the associations were novel (at least according to them), and several examples of interesting novel loci are presented. They analysed patterns of pleiotropy with a newly developed framework and identified potential effector genes using machine learning. They suggest that their results can guide, e.g., the selection of drug targets, and indicate VEGFA as a possible modulator of coronary artery disease.

The manuscript represents an important effort, is mostly well written (although maybe too complicated and detailed at some paragraphs) and provides novel aspects into disease biology. Most statistical analyses seem to be robustly conducted. I particularly appreciate that the authors give several examples of novel loci and discuss their biological importance. However, recent NMR GWA studies have had larger sample sizes. There are also some additional concerns. Please see my comments below:

Major comments:

1. The authors write: “Here we present the most comprehensive study of human metabolism to date...” However, a larger GWAS of NMR metabolites (using Estonian and UK Biobank data) was recently published as a preprint (Tambets et al. 2024; <https://www.medrxiv.org/content/10.1101/2024.10.15.24315557v2.full>), and the GWAS summary statistics of that study are publicly available. There is also another recent preprint describing GWAS of NMR metabolites in UKBB (van der Meer et al. 2024; <https://www.medrxiv.org/content/10.1101/2024.07.30.24311254v1>). In the light of the study by Tambets et al, the authors should probably modify the text (e.g., statements about the number of novel loci) throughout the manuscript and focus the contents of the manuscript on those aspects that were not covered in the preprint by Tambets et al (and van der Meer et al). Having said that, I would like to point out that the size of the study population should not be the only thing that matters. In particular, I wish to point out that compared to the preprints by Tambets et al and van der Meer et al, this study has at least one very important strength: Zoodsma et al give examples of associated loci throughout the manuscript and discuss the biological relevance of their findings. This is exactly what should be done, instead of just listing statistical associations (like many studies in the field do), and the authors should be acknowledged for this.

R2 response 1 *We agree with the reviewer that claims of novelty are often fluid, as shown by the two preprints that only appeared at the final stages of our work). We have now carefully revised any novelty claims to be more conservative. We like to note that we had already opted for a very conservative assessment of novelty, by not only considering previous papers using the NMR platform, but any association reported with any metabolite measure in the GWAS catalog. We have now included clear reference to the so far largest study on medRxiv (Tambets*

et al.) and report an observed good agreement for the common variants this work focuses on (p4, line 104).

2. Replication was assessed by comparing to the previous largest NMR GWA study (Karjalainen et al 2024, Nature 628). The authors state that only 24.8% of their associations reached GW significance in the previous study. What about replication considering higher p-value thresholds: $p < 10^{-5}$ (or even $p < 0.05$)? How many of the associations are replicated when comparing with the European results? What could be the reasons for non-replication, or is it just due to lower power in the previous study? Importantly, it should be discussed what the overall novelty of this study is compared to the study by Karjalainen et al.

R2 response 2 We thank the reviewer for these comments and followed the suggestion to perform more in-depth replication. We previously compared to the largest published meta-analysis on NMR metabolites (Karjalainen et al., Nature 2024) and now additionally included comparison to non-peer reviewed work (Tambets et al., medRxiv 2024). We observe very good effect size agreement comparing to both studies (median R^2 Karjalainen: 0.95, median R^2 Tambets: 0.94), and include a full comparison to both studies in Supplemental Figure 3. The statistical power to replicate associations in Tambets et al is higher compared to Karjalainen et al is much improved caused by their more comparable sample size and statistical power (619K vs 136K individuals).

We have further expanded our discussion on novel aspects explored in our study (p23, line 618-634), including: i) first time exploration of rare WES mQTLs at this scale and power, ii) integration of common-to-rare genetic variation to identify metabolic regulators, iii) Systematic quantification of possible sex-differences in mQTLs, iv) Development of a framework to classify pleiotropy, and v) establishment of a framework to identify putative targets for pharmacological modulation by demonstrating convergence of level and locus effects.

3. Some of the Mendelian randomization analyses are likely to be confounded by pleiotropy (e.g. Fig 6A). The authors have included SNPs from all associated loci in the MR analyses. However, most of the SNPs associated with the lipid and lipoprotein traits are very pleiotropic, and therefore the results of the MR analyses are not reliable. The authors write that they have also used lead credible sets restricted for molecular pleiotropy, but it is not clear how they accounted for pleiotropy, and in which MR analyses these were used.

R2 response 3 We apologise for the lack of clarity in our MR approach and agree with the reviewer that specificity of instruments for such highly correlated exposure is crucial. All MRs presented in the main manuscript do only use instruments with evidence for metabolite-specificity according to the criteria established in the pleiotropy section. We further implemented MR-Egger to test for pleiotropy with respect to the outcome measures and rigorously filtered results that still showed evidence for pleiotropy or heterogeneity. Only MR results passing all those filters are finally presented in the manuscript.

4. In my opinion, too much focus is given to the rare variant findings. I would suggest shortening those parts.

R2 response 4 We followed the recommendation of this and reviewer two and substantially shortened the rare variant findings (p14 – p15).

5. The paragraph “Risk mitigation of atherosclerotic cardiovascular disease beyond LDL-cholesterol” is hard to follow. Discussion and results related to VEGFA as a potential target of medication appear to be quite speculative (e.g., Fig 6), based on the current findings, if the authors are suggesting this mainly based on their suggestion of HDL particle size being causally related to CAD risk (?). Majority of research is not indicating HDL-related pathways causal for CAD, so such finding from MR analysis should be interpreted with caution.

R2 response 5 *We agree with the reviewer that these findings can be considered speculative and have now revised the corresponding section accordingly (p18, line 487). More work is needed to understand the triangle of HDL-particle size/composition and potential VEGFA-mediated uptake into the endothelium as demonstrated by independent work. We note, however, that our workflow closely resembles established drug targets such as PCSK9 and that we clearly acknowledge that not HDL-cholesterol (the target of most medications developed) is likely the causal pathway but rather other aspects of HDL particle composition.*

Minor comments:

1. Abstract: Please state the number of metabolites analyzed. Please give some examples of novel loci detected.

R2 response 6 *We added the number of associated metabolites to the abstract and included brief references to so far unknown loci.*

2. Fig 1A is very hard to read. Please consider alternative ways to summarize your findings. Maybe include Fig1B-D in Supplement.

R2 response 7 *We appreciate that not much detail can be derived from Fig. 1a but argue that it conveys several important messages not easily demonstrated by other means. For example, extensive pleiotropy at certain loci and the ability to judge mGWAS findings across different classes of metabolites. We followed the reviewer’s suggestion to make the figure more readable and enlarged the text / legend labels, but consider ancestral diversity is one of the key features of our work, and all reviewers have even requested more details, and we hence consider Fig. 1c-d essential for the main text.*

3. Fig. 2 could be in the supplement.

R2 response 8 *We have now moved Figure 2 to the supplement to shorten the manuscript.*

4. Suppl Table 1: please organize the metabolite traits in a logical way.

R2 response 9 *We have now organised metabolic traits by their biological classifications and following the order they appear in Fig 1A (see Reviewer #1 response 5).*

5. Could the authors comment on how their causal/effector gene annotations (Suppl Tables 2, 4 and/or 6) correspond to those indicated in the previous largest NMR GWA study (Karjalainen et al 2024, Nature 628). It would be good to include at least some examples/comparisons.

R2 response 10 *We followed the helpful suggestion by this reviewer and observed convergent top effector gene assignments for half of the loci (139 out of 283; 191 when considering top three; encompassing 8,025 variant – metabolite associations) that were previously biologically annotated by Karjalainen et al. and mapped to the likely same signal in our study ($r^2 > 0.5$). Convergent gene assignments had consistently higher ML-derived scores (median: 2.01 vs 1.66) providing further confidence in ML-based assignments.*

For 16 loci gene assignment was discordant (gene score >2 but no match among top 5 effector genes for biologically prioritised genes), including a locus on 19q13.11 (rs62102718) for which we prioritised PEDP with high confidence (score=2.42) as opposed to CEBPA assigned by Karjalainen et al. PEDP encodes peptidase D, which promotes adipose tissue fibrosis and insulin resistance in mice knockout models (Pellegrinelli et al. 2022 Nature Metabolism). This is a plausible explanation for the pleiotropic effects of the variant on diverse lipoprotein characteristics (n=31 metabolic traits) and supports our effector gene assignment.

We have now added this information to the manuscript (p10, line 270) and include a full comparison of our effector gene assignment to results by Karjalainen and colleagues in Supplementary Table 6.

6. The paragraph describing the results of pleiotropy analyses would benefit from shortening. The same applies to the paragraph describing the convergence of common and rare variation.
R2 response 11 *We have now shortened both sections accordingly to improve readability.*

7. Fig 4 is difficult to follow. For example, what does a single dot in the scatter plots a-d represent? In the upper part of Fig 4e, are only selected locus names shown? Fig. 4f and 4g would better be put into supplement. Maybe make Fig 4e clearer (for example, make the lines thicker if possible) and move other parts of the figure to supplement?

R2 response 12 *We followed the helpful suggestions by the reviewer and revised Fig 4 accordingly. Most importantly, we now provide better guidance on how to interpret Fig. 4a-d and improved visibility of Fig. 4e. We argue to keep figures 4f and 4g in the main manuscript as they highlight important insights: 1) the large spread in the number of associated GWAS catalog traits across different mQTL variants, and 2) the specific enrichment among GWAS catalog categories pointing to systemic effects as an explanation for pleiotropic mQTLs.*

8. Metabolite measurements in Fig 5b should be organized in a logical way. Are the metabolite ratios/percentages the most relevant traits to show? Maybe concentrate on the other measurements. Maybe put Fig 5e and 5f into supplement?

R2 response 13 *In response to this reviewer's comment, and other comments we have restructured the presentation of rare variant findings along with possible common-to-rare conversion. As a result, display items changed (e.g., previous Fig. 5e-f have now been omitted), but we incorporated the helpful comments from the referee.*

9. Fig 6 – what does it mean to be a “not-LDL associated locus”? How is this defined? Maybe by comparison to largest available LDL cholesterol GWAS? Fig 6D does not give any information as such. Maybe present in a different way, or pick examples of most relevant associations?

R2 response 14 *We rephrased the corresponding section accordingly, to clearly state that associations with LDL-cholesterol were defined based on the current cohort.*

10. Could locuszoom plots be shown for exemplary loci with interesting colocalizing metabolite and trait associations?

R2 response 15 *We have created locus zoom plots for all colocalization and provide them with the GitHub for further reference.*

11. Please discuss the pros and cons of using a single cohort in this kind of GWAS.

R2 response 16 *We now include a brief discussion about the pros and cons of performing mGWAS in a single large cohort compared to meta-analysing multiple cohorts (p23, line623).*

Reviewer #3 (Remarks to the Author): This manuscript reports on the genetic architecture of metabolomic traits as quantified by H1-NMR (Nightingale platform) among approximately 450,000 participants of the UK Biobank (UKB) of British White European (EUR), British African (BA) and British Central South Asian (BSA) ancestries. The authors provide a comprehensive catalogue of approximately 30,000 locus-metabolite signals mapping to 753 regions, with effector gene assignment, characterization of pleiotropy, establishment of allelic series, and assessment of how these findings relate to human disease. The detected common variant associations show largely consistent effect sizes across ancestries and across sexes. This group of authors is very experienced with genome-wide screens of intermediate molecular traits such as metabolites or proteins. This is also evidenced by their group obtaining early access to the UKB Nightingale measurements. The analyses are state-of-the-art, and the authors did an excellent job on the thorough exploration of the metabolome, phenome and genome. The manuscript is very well written, with nice graphics. I do not have any substantial general concerns about the presented findings, although it is somewhat of a pity that - despite the abundance of significant results - many findings are expected and confirmatory and few insights directly translate into altered disease risk, or – if they are unexpected and novel – it is difficult to know if they really represent new causal relationships (see below). However, this is not the authors' fault, and the presented findings represent a robust and very valuable resource for the scientific community.

The authors are asked to address the comments below, with major points related to the choice of showcases and the amount of speculation in the results and discussion section.

Major:

1. Figure 2c: although there are many fewer variants in the credible set (CS) in multi-ancestry finemapping, the PIP of the best variant in EUR is not much lower than the PIP in multi-ancestry finemapping. Importantly, it is also just as centered on the PTRF gene, which also happens to be the closest gene anyway. Could the authors highlight another example that better illustrates improved resolution of the most likely causal gene owed to multi-ancestry finemapping?

R3 response 1 *The reviewer is right, that this is not the best example to demonstrate the benefits of trans-ancestral fine mapping. This is, in part, a result of the methodological shortcomings and missing scale in non-European ancestral groups. While we used one of the most recently developed methods, poor coverage of SNPs across ancestries and already very narrow European credible sets (because of the large sample size) have limited our ability to demonstrate more staggering improvements. We do think, however, that demonstrating refinement of credible set in massively European-dominated setting is still important to exemplify the value of even moderately sized non-European cohorts. Also, in response to concerns raised by the other reviewers, we have now toned this section down substantially to conclude that while large-scale European-centric efforts can still benefit from non-European contributions, the real value would be in WGS data in equally sized cohorts.*

2. Machine-learning effector gene assignment: although the authors report statistics in favor of their RF classifiers, the proportion of loci in which a significant gene-burden finding supports the assigned prioritized gene is disappointingly low (not even a third of them is among the top three ML-based assigned effector genes). How do the authors explain this? Given the evidence through human support via pLoF variants, wouldn't it be more straightforward to assign the gene implicated by the burden test as the causal gene rather than relying on an algorithm that includes evidence from pervasive eQTLs etc.? How do the enrichment statistics reported throughout the paper compare when these analyses are repeated with the genes supported by significant gene burden tests prioritized as effector genes?

R3 response 2 *We agree with the reviewer, that evidence from pLoF variants can greatly enhance confidence in candidate gene assignment. As part of the review process, we have now restructured the presentation of common-to-rare effect convergence, presenting allelic series for overlap between rare/burden analysis and any ML-prioritized gene (Supplementary Table 6). The still somewhat comparatively low number (55.5%) might be best explained by the biologically informed selection of true-positive sets, being blind to any new, more distal discoveries, and the low informative content of some predictors, including eQTLs (that ranked low).*

However, we'd like to emphasize that most common variants (92.3%) were not localised near a rare variant and hence the need to develop scalable strategies to enable effector gene prioritisation without supporting evidence from pLOFs. We now also provide additional validation of effector gene assignment (p9, line 262) but also clearly acknowledge the need for improvements.

Given the rather small number of common-to-rare pairings, tissue enrichment analyses were unaffected, like when using the closest gene when running this enrichment analyses.

3. Lines 407-410: the authors speculate that the association between pLoF variants in SLC30A10 and lipoprotein particle composition may be mediated by manganese (Mn). Can they investigate whether the same variants driving the association with NMR particles are also related to blood Mn levels, using either GWAS data such as PMID: 38594418, or biobank sequencing data in case Mn measurements are available? If unavailable, the authors can test whether mutations in SLC30A10 known to cause the monogenic condition Hypermanganesemia With Dystonia 1 are present (at least in the heterozygous state) in the UKB and whether carrier have changes in the implicated metabolites in the expected direction.

R3 response 3 *We followed the reviewer and observed that the common variant we identified (rs11118310) at this locus is not related to the nearby signal associated with Mn levels (rs1776029) in previous studies ($r^2=0.07$; Ng et al. HMG 2015). Considering this missing crucial link, we omitted the presentation of the example entirely. As part of our revised strategy to integrate rare variant findings, we observed that even among allelic series in well-described metabolic genes, phenotypic associations differed, and we now present these as examples (p15).*

4. Lines 494-508. I cannot follow the authors' logic around the SLC22A2 and creatinine/CKD showcase. The authors found that pLOF variants in the gene known to encode a creatinine transport protein, SLC22A2, are associated with creatinine levels and with chronic kidney

disease, which is commonly defined based on the glomerular filtration rate estimated from creatinine. The variants are not associated with an alternative and superior marker of kidney function, serum cystatin C. The most likely explanation then is that the detected variants lead to differences in creatinine metabolism (and kidney disease defined from creatinine), but not truly to kidney dysfunction or kidney disease. If these variants were associated with kidney disease, then there would be an association with cystatin C levels. Cystatin C is available in the UKB, and the authors can and should test the associations of their rare variant masks with cystatin C levels. If it turns out that this finding mainly represents an effect on creatinine metabolism, then it is not the best showcase for the abstract, a main figure, and a full paragraph. Others have anyway reported rare LoF variants in this gene in association with creatinine-based kidney function in the UKB before (PMID: 36890159).

R3 response 4 *We have now performed the analysis requested by the reviewer. The results supported the missing similarly strong association with cystatin C measurements ($p=0.0004$). In light of this, we have now omitted this example from the revised manuscript.*

5. Lines 530-552: the authors speculate on the potential of activating VEGFA to influence properties of HDL particles as a potential mechanism to reduce CAD risk. However, the manuscript cannot establish that these properties actually mediate CAD risk and are not only proxies for unmeasured molecules, nor that one of the other lipoprotein properties that colocalize with CAD risk at VEGFA such as LDL cholesterol and triglyceride properties (Suppl Table 13) are the ones that are important. Further, MR results in ST 12 suggest that HDL cholesterol levels show just as strong potentially causal associations with CAD risk as do other HDL particle properties. Moreover, VEGFA is a potent angiogenesis factor that is activated in many cancers, raising questions if it is a worthwhile target to try to activate. I am not convinced that this is the strongest example to highlight in the abstract and a full paragraph of the results and in a main figure and suggest replacing it or toning it down.

R3 response 5 *We appreciate, that our previous presentation of the results had been insufficient and revised the section accordingly (p19, line 510). The main findings are: 1) there is evidence that different levels of HDL particle characteristics are associated in a protective fashion with CAD risk, although we totally agree with the reviewer, that we cannot for sure say which, and 2) we obtained evidence that genetic variation at the VEGFA locus seems to change exactly those with almost no effect on LDL-cholesterol levels ($p=0.2$). Most importantly, pharmacological modulation of VEGFA further supports an HDL-specific effect at the disease site of CAD (Velagapudi et al. 2017 Arterioscler. Thromb. Vasc. Biol.) and has recently been shown to improve healthy lifespan in mice (Grunewald et al. 2021 Science) that may at least hint at the potential of intervention.*

6. Lines 589-595: the last two sentences are redundant, or something is missing. More importantly, if the implicated metabolite only proxies a disease state such as chronic inflammation, then how much can we rely on MR results that implicate potentially “causal” associations? If in addition we do not know how this metabolite relates to type 2 diabetes, again this is too much speculation for a results section of a manuscript.

R3 response 6 *We agree and have now omitted any such mechanistic speculation.*

7. Discussion, lines 628-631: true, but the authors do not know whether the NMR-based properties that they study are actually the ones on which the genes primarily act. They can now be measured and hence used for genetic association studies, but experimental proofs

they represent the biologically important properties is mostly missing. I suggest removing this sentence.

R3 response 7 *We agree with the reviewer and removed this conclusion from the text rather refer to the obstacles of establishing modes of pleiotropy in general (p22, line 592).*

8. Data availability: GWAS summary statistics and the results from genome-wide burden tests need to be shared with the scientific community, i.e., made available for download or submitted to the GWAS catalogue or similar. It is not enough to say that they will be made available upon publication, it is important to know how they will be shared. The authors can set up temporary links or protected access so that reviewers can see how they are planning to share these results.

R3 response 8 *We absolutely agree with the reviewer and have now uploaded all statistics, including meta-analysis, ancestral strata, exWAS and gene burdens, to the GWAS catalog. All accession IDs are given in Supplementary Table 17.*

9. Methods: lines 819 ff. Is it correct that the joint statistics were computed in a different set of individuals (unrelated EUR) compared to the marginal Susie statistics coming from the overall sample? If so, then it does not seem to correct to compare potential attenuation of effect sizes, since the statistics stem from a partially non-overlapping sample.

R3 response 9 *The reviewer is correct that joint models were computed in a subcohort of UKB. We agree that different subsets of the population can lead to varying estimates but argue that those changes are neglectable compared to the changes seen for some loci by selecting somewhat correlated SNPs into different credible sets. This filter eliminated only a small, but relevant, proportion of findings that may otherwise lead to spurious findings.*

Minor:

1. The authors compare effect sizes between EUR and BA or BSA, respectively (Fig 1c,d). Can they please also compare the results from the trans-ethnic meta-analysis and the EUR subset, for all trans-ethnic sentinel variants? In panel 1c, what is the subgroup of dots close to the horizontal line – do they all come from one or few very pleiotropic loci, giving rise to many dots that do not align?

R3 response 10 *We have followed the recommendation of the reviewer and now additionally provide a comparison of findings between the trans-ethnic MA and EUR only results (Figure 2). UKB and our meta-analysis is dominated by European individuals (435K Europeans vs 9K British-Asians vs 7K British-Africans), and it is therefore not surprising to see that the meta-analysis effect sizes correlate very well to the European-only results.*

We further investigated variants strongly associated in European individuals but without evidence in British African individuals (e.g. variants in Fig. 1C close to the horizontal line). These snp:trait associations (Figure 2, left panel) can be traced back to 95 unique variants, originating from at least 36 unique genetic regions. As Figure 2 (right panel) shows, the allele frequencies of variants significant in either population correlate well, and this includes variants with strong evidence in Europeans but not British Africans (black dots). We do not see examples of strongly associated variants with largely differing allele frequencies but note that this analysis excludes variants totally private to either population.

Figure 2. Comparison of effect sizes between the trans-ethnic meta-analysis (x axis) compared to European-only derived effect sizes (y axis).

Figure 3. Comparison of allele frequencies from loci strongly associated in Europeans but not associated in British African individuals. The left figure is a replica of Fig 1C in the manuscript, comparing estimated effect sizes between Europeans and British Africans. The black dots are strongly associated in Europeans ($\beta < -0.05$ or $\beta > 0.05$) but not in British Africans ($\beta < 0.025$ and $\beta > -0.025$). The right figure shows the allele frequencies of the same variants between Europeans (x) and British Africans (y). Black dots are like the left figure.

2. Lines 171-173: how do the authors explain this substantial increase in the number of high-confidence variants upon multi-ancestry finemapping? Were these variants members of the credible sets in EUR already (should be, since these are larger) and only had lower PIPs?

R3 response 11 *The reviewer is right. The increase in high-confidence variants is due to crucial refinement of EUR credible sets based on differential haplotypes in BA and/or BSA participants. Only variants already prioritised in EUR people entered the trans-ancestral fine mapping.*

3. Lines 239-241: please include the discovery trait in the GLGC in the text. Presumably it is LDL cholesterol levels, but it is not clear from the text. If so, how do the authors explain that have completely insignificant p-values in their analysis: is it only the smaller sample size? Please include the betas, not just the p-values.

R3 response 12 *We have added the requested information (p8, line 220). Briefly, while the smaller sample size compared to the GLGC may account for the null finding for LDL-c in our study, the much stronger association with the same discovery power for (V)LDL particle size suggests that rs738409 acts primarily via this route on LDL-c.*

4. Methods: Finemapping: if I did not miss it, please add which reference panels were used for finemapping in BA and BSA?

R3 response 13. *We clarified, that in-sample reference panels have been used for BA and BSA participants (paragraph: 'multi-ancestry finemapping').*

5. Lines 420-23: a potential role of NDRG2 in histidine metabolism is highly speculative at this point, without any experimental evidence, even in the literature. It would be better to replicate this association when highlighting it in the manuscript. It can happen that metabolites are mis-annotated from a specific method such as NMR. If other associations in this work support the correct annotation of histidine (e.g., with variants in HAL, I didn't check), then I would mention this in this context.

R3 response 14 *We agree with the reviewer that the finding is highly speculative, but at least the common variant close to NDRG2 has been identified in previous work (Kettunen et al. 2016 Nat Comms), and strong associations with rare exonic variants in HAL further corroborate correct assignment of HAL. We note that histidine was among the most technically variable metabolites measured on the NMR panel (Figure 2B, left panel, Ritchie et al, Scientific Data, 2023 (<https://doi.org/10.1038/s41597-023-01949-y>)) and we cannot completely rule out that the associations with common and rare variants mapping to NDRG2 possible represent unspecific effects. We therefore omitted in-depth presentation of this example in the revised manuscript.*

6. Figure 5d is a great panel. The authors should be commended on integrating common and rare variant evidence.

R3 response 15 *We thank the reviewer for this positive assessment of our work!*

Reviewer comments

Reviewer #1 (Remarks to the Author): The authors have answered all my points. The manuscript is improved.

Thank you for your time and positive assessment of our work.

Reviewer #2 (Remarks to the Author): The manuscript has clearly been improved after revision.

Thank you for your considerations, we are glad you find the manuscript improved and ready for publication.

Reviewer #3 (Remarks to the Author): The authors have thoroughly revised their manuscript and addressed many of my initial comments. Especially comments related to biological plausibility of speculative nature or showcases of loci not supported by sufficient evidence have been revised (mostly by removal). Important additions are a comparison of findings to those from other studies, and a significantly revised and (in my opinion) improved workflow to assign effector genes by considering both common and rare variant support for a given gene, and the sharing of all summary statistics supporting the results.

We are glad to hear that the reviewer finds the manuscript and specifically the effector gene assignment workflow improved after considering the earlier suggestions.

However, several aspects require further attention and potentially modification:

Major:

- Newly added showcase on prioritization of *PEPD* as the effector gene (p.9, line 268 ff): a look at Suppl Table 6 shows that for the locus tagged by rs62102718, there was only one candidate gene (col H) in the region anyway, whereas metabolite associations tagged by other SNPs in this locus also list *CEBPA* as a potential candidate gene. If there is only one candidate and the lead variant is sitting in an intron of the assigned gene, then is this the best showcase to support the value of this prioritization workflow?

R3 response 1. We thank the reviewer for the thoughtful revision of the newly added example and appreciate that more explanation is needed. The reason why *PEPD* is listed as the single effector gene at this locus is due to the strong evidence from our prioritization algorithm which estimates the next best gene (*CEBPG*) to be almost half as likely (gene score: 1.4, compared to 2.4 for *PEPD*). This assignment is also corroborated by a recent mouse model (Pellegrianni et al., Nat Metabolism 2022). We have now added an additional explanation to the method section to clarify the details of how the algorithm prioritizes effector genes at a given locus (p30, l914-917).

- New JAK2 example (p.14): the authors could improve their presentation of this finding by searching for more literature supporting a role of JAK2 in lipid and hepatic metabolism. A quick search shows that such literature may be published, in addition to the mentions CHIP connection (e.g., <https://pubmed.ncbi.nlm.nih.gov/23782652/>).

R3 response 2. We appreciate that our previous interpretation of the *JAK2* results was too narrow-minded and have now expanded upon a potential role of *JAK2* variants in hepatic and peripheral lipid metabolism, including evidence from mouse models (PMIDS: 23782652, 28724798) as well as human studies (PMIDS: 29083408) (p14, l382-392).

- New/revised Figure 5: this is interesting. For the regression estimates presented in panels e and f, more information would help. For instance, was there any overlap in carriers of the different variants? If so, then conditionally independent effect estimates should be presented, which may turn out important especially when effect directions can differ across variants.

R3 response 3. We followed the recommendation and identified no overlap in rare variant carriers for the presented rare variants and further highlight, that all presented rare variant findings had to pass statistical correction for common alleles, including preservation of effect estimates.

- Limitations, p.23: it is an important limitation that NMR measurements of apolipoproteins can be affected in the presence of rare genetic variants. Given the focus on rare variants in the *APOA1* gene in the main MS, in particular in Figure 5, their association with apolipoprotein A1 levels and other NMR metabolites, and the availability of apolipoprotein A measurements by classical blood biochemistry in the UKB, the authors should support these findings by using the blood biochemistry measurements in a sensitivity analysis. Please also show the correlation between “standard” and NMR apoA1 levels.

R3 response 4 We have now implemented this extremely helpful suggestion and added effect estimates for clinically measured ApoA1 levels to the figure for comparison. While we noted minor differences, our findings and overall conclusion remained unaffected. We have now added mentioning of the caveat that for some rare alleles, quantification of ApoA1 by ¹H-NMR spectroscopy may need recalibration (p16, l438). In general, ApoA1 as measured by immunoturbidimetric analysis and ¹H-NMR spectroscopy correlated well ($r=0.88$; Supplementary Figure 14).

Minor:

- Figure 2 has been removed from the main MS, subsequent figures need to be renumbered.

R3 response 5 We apologize for this oversight and renumbered all figures accordingly in the revised version of the manuscript.

- Please include in the main text that all regions found in non-EUR groups, other than the

one at CD36 identified exclusively identified in BA participants, were also observed in the EUR participants.

R3 response 6 We agree with the reviewer and have now added this (p4, l119-123)